# ZFYVE21 is a complement-induced Rab5 effector that activates non-canonical NF-κB via phosphoinosotide remodeling of endosomes

Caodi Fang[1], Thomas D. Manes[2], Lufang Liu[1], Kevin Liu [1], Lingfeng Qin[3], Guangxin Li[3], Zuzana Tobiasova[2], Nancy C. Kirkiles-Smith[2], Manal Patel [4], Jonathan Merola[3], Whitney Fu [1], Rebecca Liu[2], Catherine Xie[2], Gregory T. Tietjen[3], Peter A. Nigrovic[5], George Tellides[3], Jordan S. Pober[2] & Dan Jane-wit[1]

Complement promotes vascular inflammation in transplant organ rejection and connective tissue diseases. Here we identify ZFYVE21 as a complement-induced Rab5 effector that induces non-canonical NF-κB in endothelial cells (EC). In response to membrane attack complexes (MAC), ZFYVE21 is post-translationally stabilized on MAC+Rab5+ endosomes in a Rab5- and PI(3)P-dependent manner. ZFYVE21 promotes SMURF2-mediated poly-ubiquitinylation and proteasome-dependent degradation of endosome-associated PTEN to induce vesicular enrichment of PI(3,4,5)P3 and sequential recruitment of activated Akt and NF-κB-inducing kinase (NIK). Pharmacologic alteration of cellular phosphoinositide content with miltefosine reduces ZFYVE21 induction, EC activation, and allograft vasculopathy in a humanized mouse model. ZFYVE21 induction distinctly occurs in response to MAC and is detected in human renal and synovial tissues. Our data identifies ZFYVE21 as a Rab5 effector, defines a Rab5-ZFYVE21-SMURF2-pAkt axis by which it mediates EC activation, and demonstrates a role for this pathway in complement-mediated conditions.

[1] Division of Cardiovascular Medicine, Yale University School of Medicine, New Haven, CT 06520, USA. [2] Department of Immunobiology, Yale University School of Medicine, New Haven, CT 06520, USA. [3] Department of Surgery, Yale University School of Medicine, New Haven, CT 06520, USA. [4] St. John's College, University of Cambridge, Cambridge CB2 1TP, UK. [5] Division of Rheumatology, Immunology and Allergy, Brigham and Women's Hospital and Division of Immunology, Boston Children's Hospital, Boston, MA 02115, USA. Correspondence and requests for materials should be addressed to D.J-w. (email: dan.jane-wit@yale.edu)

Endothelial cell (EC) activation regulates tissue inflammation in solid organ transplant rejection and connective tissue diseases. We previously showed that complement membrane attack complexes (MACs) trigger EC activation[1,2]. MACs are transmembranous structures formed from complement components that intercalate into target cell membranes to induce various non-cytolytic responses. We[1,2] and others[3,4] have examined non-lytic and stimulatory properties of MAC. To do this, we used sera from allo-sensitized transplant candidates containing high-titer panel reactive antibody (PRA) to induce alloantibody-mediated MACs on human EC in vitro[1,2] and in vivo[5]. In these models, MAC and not IgG binding or anaphylatoxins elicited EC activation by inducing via posttranslational stabilization of nuclear factor (NF)-κB-inducing kinase (NIK)[2], to activate non-canonical NF-κB.

NIK stabilization occurs in response to ligand:receptor interactions involving tumor necrosis factor receptor (TNFR) super-family members such as CD40 and LT-βR[6]. Basally, NIK is transcribed, translated, and captured by a degradation complex containing TRAF3 and cIAP2 that targets NIK to the proteasome. Following receptor-mediated activation, the degradation complex is recruited to the activating receptor where TRAF3 is degraded, allowing NIK to slowly accumulate over ~12–24 h. Informed by a genome-wide small interfering RNA (siRNA) screen[1], MAC were shown to initiate an alternative, TRAF3-independent mechanism of NIK stabilization in ~30 min. In this alternative mechanism, membrane-bound MAC are internalized via clathrin-mediated endocytosis and transferred to Rab5+ endosomes to form a MAC+Rab5+ compartment. In a process dependent upon clathrin-mediated endocytosis of MAC as well as activated (GTP-bound) Rab5, this compartment recruits activated (phosphorylated) Akt to form a pAkt+Rab5+ endosome. The modified endosome then sequesters NIK from the TRAF3/cIAP2 degradation complex, allowing it to initiate non-canonical NF-κB.

Our observations raised questions as to the characteristic feature(s) of MAC+Rab5+ endosomes that permitted recruitment of pAkt and NIK. Rab5 is a small GTPase that mediates downstream functions by recruiting effector proteins. Here ZFYVE21 is identified as a posttranslationally induced Rab5 effector that triggers phosphoinositide (PI) remodeling of early endosome membranes to mediate non-canonical NF-κB signal activation and tissue inflammation. ZFYVE21 is further explored both as a therapeutic target to modulate EC activation in vivo and as a specific biomarker for complement-mediated signaling in human tissues.

## Results

**MAC-induced and endosome-dependent stabilization of ZFYVE21.** To investigate how MAC+Rab5+ endosomes recruit pAkt, we performed proteomic analyses of FACS (fluorescence-activated cell sorting)-sorted MAC+Rab5+ vesicles. Human umbilical vein endothelial cells (HUVECs) were stably transduced with a Rab5-GFP fusion protein and treated with "high" PRA sera supplemented with AF647-labeled C9, the most abundant complement component in MAC. AF647-C9 supplemented PRA sera allowed formation of dually fluorescent MAC+Rab5+ vesicles. ECs were then sonicated and the double positive Rab5-GFP+C9 AF647+ (MAC+Rab5+) population was gated and sorted (Supplementary Fig. 1a). FACS-sorted, single positive Rab5-GFP+ vesicles in vehicle-treated ECs were used as controls. For our proteomic analyses, we gated on smaller event sizes to exclude membrane-bound organelles like endoplasmic reticulum, nuclei, and mitochondria, which may have contained docked Rab5+ vesicles.

Prior to analysis, we performed immune-electron microscopy experiments to analyze C9+Rab5+ vesicles prior to and following FACS isolation. Intact C9+Rab5+ vesicles in native HUVECs prior to sorting showed an average diameter of 1062 ± 530 nm vs 670 ± 236 nm in control Rab5+ vesicles (Supplementary Fig. 1b). Following FACS sorting, vesicles were morphologically intact (Supplementary Fig. 1c, left) and showed Gaussian size distributions (Supplementary Fig. 1c, right top). C9+Rab5+ vesicles had an average diameter of 112 ± 20 nm (Supplementary Fig. 1c, right bottom) vs 37 ± 7 nm in control Rab5+ vesicles[7].

C9 has strong propensity to form aggregates and as such FACS-sorted C9+Rab5+ events may have reflected structures containing aggregates of C9 rather than MAC. To test for C9 AF647 aggregation, we tested absorbance of C9 AF647 prior to and following ultracentrifugation of the protein product based on the reasoning that, if significant protein aggregation occurred, we would expect the measured absorbance of C9 AF647 supernatants to decrease following ultracentrifugation. We tested two separate batches of C9 AF647 along with a commercial goat antimouse IgG AF647 antibody (Ab) as a labeled protein control. Our data showed that the absorbance of the supernatants from two separate batches of C9 AF647 declined by ≤5% after ultracentrifugation, which was less than that seen in the control Ab (Supplementary Fig. 1d, left, middle). Changes in absorbance after ultracentrifugation did not differ between C9 AF647 batches and were not significantly different than changes in measured absorbance in phosphate-buffered saline (PBS) controls (Supplementary Fig. 1d, right). We subsequently tested the ability of ultracentrifugation products of C9 AF647 to be internalized into C9+Rab5+ endosomes. C9 AF647 supernatants prior to (C9 AF647 Pre) and following (C9 AF647 Post) centrifugation were used to treat HUVECs under complement-activating conditions. Our data showed that C9 AF647 Pre and C9 AF647 Post supernatants did not significantly differ in their ability to generate C9+Rab5+ vesicles in HUVECs (Supplementary Fig. 1e, bottom graph). Moreover, both C9 AF647 Pre and C9 AF647 Post products yielded significantly fewer C9+Rab5+ vesicles per cell when incubated with the cells compared to uptake of the same labeled proteins when combined with PRA sera. Together, these data indicated that C9 AF647 aggregates minimally contributed to the overall numbers of FACS-sorted C9+Rab5+ vesicles used for downstream proteomic analyses.

Sorted vesicles were subsequently subjected to trypsin digestion and label-free proteomic analysis by mass spectrometry–liquid chromatography. Three independent experiments uncovered 5450 individual proteins. Of these, 440 proteins were exclusive to MAC+Rab5+ vesicles, 688 proteins were exclusive to control Rab5+ vesicles, and 4322 proteins were shared between groups (Supplementary Fig. 1f). As expected, proteomes of MAC+Rab5+ endosomes showed immunoglobulin heavy and light chains as well as complement components 1–9. Additionally, in accordance with prior findings[1], MAC+Rab5+ proteomes contained Akt, NIK, and p100 as well as components of clathrin-mediated endocytosis including clathrin, dynamin-2, and AP2.

As pathway-activating signalosomes form on MAC+Rab5+ endosomes and depend upon GTP-Rab5[1], we hypothesized that this compartment would be enriched in Rab5 effectors, which, by definition, bind to GTP-Rab5 and mediate a downstream response[8]. We therefore searched vesicular proteomes for proteins containing a FYVE domain, which is a PI(3)P lipid-binding motif[9–11] present in Rab5 effectors[8]. Based on this, we identified three proteins, ZFYVE7, ZFYVE17, and ZFYVE21, and of these, only ZFYVE21 spectral counts were detected at significantly higher quantity in MAC+Rab5+ vesicles compared to control Rab5+ vesicles (Supplementary Fig. 1g). Total cellular ZFYVE21 protein was low basally but was rapidly induced within minutes following PRA treatment (Fig. 1a). This rapid induction was posttranslationally mediated as treatment with a proteasome

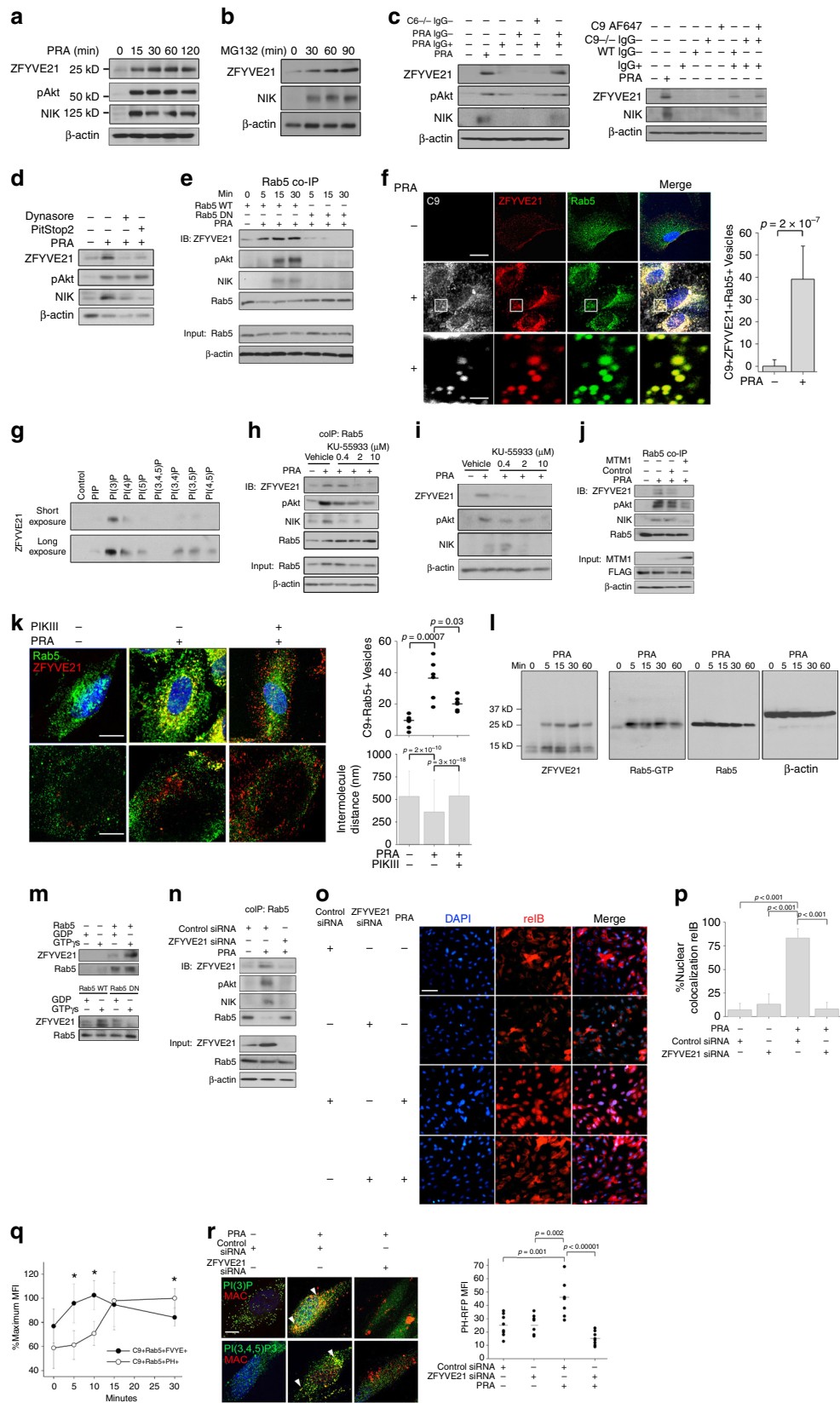

inhibitor, MG132, induced ZFYVE21 as well as NIK, a protein known to be posttranslationally stabilized (Fig. 1b). We next interrogated complement-related activation product(s) required for ZFYVE21 induction in sera fractionation experiments. We found that only the IgG+ portion of PRA combined with the IgG− fraction from normal human sera could induce ZFYVE21 and NIK, whereas the IgG+ portion of PRA sera alone or the IgG+ portion of PRA sera plus the IgG− fraction from C6-deficient reference sera could not (Fig. 1c, left). The ability of IgG+ fractions to induce pAkt is consistent with prior studies showing

**Fig. 1** Membrane attack complex-induced and endosome-dependent stabilization of ZFYVE21. Human umbilical vein endothelial cells (HUVECs) were treated with panel reactive antibody (PRA) sera (25% v/v in gelatin veronal buffer) (**a**). HUVECs were treated with MG132 (25 μM) (**b**). HUVECs were treated with IgG+ and IgG− sera fractions from PRA and C6-deficient sera (50% v/v in gelatin veronal buffer) for 30 min prior to western blotting (**c**, left). HUVECs were treated with IgG+ and IgG− sera fractions from PRA sera, IgG− sera fractions from C9-deficient sera, C9 AF647, or in combination for 30 min (**c**, right). HUVECs were pretreated with Dynasore (80 μM) or Pitstop2 (30 μM) for 30 min, **d**. HUVECs stably transduced with Rab5 wild type (WT) or Rab5 dominant negative (DN) (S43N) were treated with PRA prior to co-immunoprecipitation of Rab5 (**e**). Three-color confocal microscopy of PRA-treated (**f**, top scale bar 8 μm, bottom scale bar 530 nm). Recombinant ZFYVE21 (1 μg) was co-incubated with lipid-coated beads and probed by Western blot (**g**). HUVECs were treated for 30 min with KU-55933 prior to blotting of Rab5 co-immunoprecipitates (**h**) and whole-cell lysates (**i**). HUVECs stably transduced with control or MTM1-expressing vectors were treated with PRA sera for 30 min prior to Rab5 co-immunoprecipitation (**j**). Rab5+ZFYVE21+ endosomes were quantified by confocal microscopy (5 nM, **k**, top row, scale bar: 10 μm). Intermolecule distances between Rab5 and ZFYVE21 were calculated using super-resolution microscopy (**k**, bottom row, scale bar: 10 μm). Myc-tagged ZFYVE21 was used in far western blots (**l**, left blot). Far western blot membranes were stripped and probed by western blot (**l**, right three blots). Recombinant Rab5 (**m**, top), Rab5 WT, or Rab5 DN (**m**, bottom) were incubated with GDP or GTPγS prior to addition of ZFYVE21. HUVECs transfected with control or ZFYVE21 siRNA (**n**). HUVECs transfected with control of ZFYVE21 siRNA were treated with PRA for 4 h prior to immunofluorescence (I.F.) analysis (**o**, **p**, scale bar: 85 μm). Lipid reporter mean fluorescent intensity PI(3)P:PI(3,4,5)P3 ratios gated on C9+Rab5+ vesicles were calculated (**q**, **r**, scale bar: 10 μm). *$p < 0.05$, analysis of variance. For I.F., ≥3 fields were analyzed per group per experiment, and each experiment was repeated two times. For confocal microscopy analyses, ≥10 individual cells were analyzed per group, and each experiment was repeated four times. Super-resolution microscopy was repeated twice. All western blot assays were conducted two to four times. Representative data shown

this effect with an anti-major histocompatibility complex (anti-MHC) Ab[12]. However, the concurrent inability of the IgG+ fraction alone to induce ZFYVE21 and NIK indicated that IgG binding to ECs activated functionally distinct pools of Akt. To strengthen these data, we repeated these studies with fractions of PRA sera combined with C9-deficient sera and/or C9 AF647 protein product used in the endosome sorting studies above. We found that C9 AF647 rescued induction of ZFYVE21 and NIK in cultures containing C9-deficient sera and that C9 AF647 alone was unable to induce these molecules (Fig. 1c, right, lane 8 vs lane 9). These data showed that ZFYVE21 induction required MAC and not alloantibody or anaphylatoxins. We next asked whether internalization of MAC was required for inducing ZFYVE21, a process required for MAC+Rab5+ endosome formation and posttranslational stabilization of NIK[1]. We pretreated HUVECs with Dynasore and Pitstop2, pharmacologic inhibitors of endocytosis, and found that these drugs blocked induction of ZFYVE21 and NIK (Fig. 1d). pAkt was not blocked by Dynasore nor Pitstop2, congruent with prior results[1]. Following (or concomitant with) its induction, ZFYVE21 along with pAkt and NIK were rapidly recruited to Rab5+ endosomes within minutes in a process requiring Rab5 activity (Fig. 1e). As our prior analysis of FACS-sorted C9+Rab5+ events yielded vesicles that were small when compared to sizes of bona fide Rab5+ endosomes in situ, we performed confocal microscopic studies to demonstrate colocalization of MAC (C9), ZFYVE21, with Rab5 endosomes (Fig. 1f, Supplementary Data 1), which, quantitatively, averaged 39 ± 15 endosomes per cell. These data collectively indicated that MAC+Rab5+ endosomes sequestered ZFYVE21 from posttranslational degradation in a Rab5-dependent manner, similar to what we previously observed for NIK[1].

FYVE domains bind to lipids, implying that vesicular sequestration of ZFYVE21 may have required lipid binding in addition to Rab5 activation. To identify candidate lipid(s) required for ZFYVE21 recruitment, we tested binding of ZFYVE21 with PI-coated beads and found that ZFYVE21 selectively bound to PI(3)P, consistent with other FYVE domain-containing proteins (Fig. 1g)[9–11]. We performed four experiments to assess a role for PI(3)P in ZFYVE21 stabilization. First, we treated EC with KU-55993 to inhibit class II and III phosphoinositide 3 kinases (PI3Ks)[13], thereby depleting cellular membranes of PI(3P). This treatment blocked ZFYVE21 recruitment to Rab5+ vesicles in co-immunoprecipitation (co-IP) experiments following PRA treatment (Fig. 1h), resulting in decreased overall cellular levels of ZFYVE21 in whole-cell lysates (Fig. 1i). We second observed

similar attenuation of vesicular sequestration of ZFYVE21 upon overexpression of myotubularin (MTM1)[14], a PI(3P) phosphatase, compared to controls (Fig. 1j). Third, we treated EC with PIKIII, a selective Vps34 inhibitor, and found that PI(3)P lipid depletion with this compound significantly blocked ZFYVE21 colocalization with Rab5+ endosomes (Fig. 1k, top row). Finally, in follow-up studies using super-resolution microscopy intermolecule distances between ZFYVE21 and Rab5 were significantly reduced upon treatment with PRA and this effect was significantly reversed with PIKIII (Fig. 1k, bottom row, Supplementary File 1).

Interestingly, PIKIII redistributed ZFYVE21 to a distinct Rab5-negative vesicular population rather than into the cytosol. Given our prior observations of reduced stability of ZFYVE21 in cells containing Rab5 dominant negative (DN) (Fig. 1e) and in cells depleted of PI(3)P (Fig. 1h–j), we surmised that ZFYVE21 was shunted to vesicles implicated in protein degradation when interactions between ZFYVE21 and Rab5 were blocked with depletion of the PI(3)P lipid-binding anchor. This prompted us to examine whether ZFYVE21 directly interacted with Rab5. In far western blots, myc-tagged ZFYVE21 protein bound to a 25-kD band in PRA-treated but not untreated controls, indicating that an inducible change in the target protein(s) for ZFYVE21 had occurred (Fig. 1l, left blot). As Rab5 effectors preferentially bind to active, GTP-bound Rab5 whose molecular weight corresponded to the 25-kD band size, we reasoned that the increase in ZFYVE21 binding was a result of PRA-induced Rab5 activation, consistent with our observation that Rab5 DN, which carries an S43N mutation preventing binding to GTP, blocked vesicular recruitment of ZFYVE21 (Fig. 1e). As an initial test of this hypothesis, we stripped and reprobed membranes with anti-Rab5 GTP Ab and found increased Rab5-GTP upon PRA treatment while total Rab5 levels remained unchanged (Fig. 1l). A direct interaction of ZFYVE21 with active Rab5 was determined by incubating recombinant Rab5 with GDP or non-hydrolyzable GTPγS to form inactive Rab5-GDP or active Rab5-GTPγS complexes, respectively, prior to incubation with recombinant ZFYVE21 protein. We observed selective ZFVYE21 binding to active Rab5-GTP but not Rab5-GDP (Fig. 1m, top), a finding corroborated using Rab5 wild-type (WT) and Rab5 DN isoforms isolated from transduced HUVECs (Fig. 1m, bottom). Finally, to define a functional role of ZFYVE21, we performed siRNA knockdowns of this molecule and found that this treatment prevented both pAkt and NIK recruitment to Rab5+ vesicles, thereby preventing stabilization of NIK (Fig. 1n). Furthermore, ZFYVE21 inhibition significantly blocked nuclear translocation of

relB (Fig. 1o, p), an NIK-dependent process indicating terminal non-canonical NF-κB activation[6]. These data identified ZFYVE21 as a Rab5 effector mediating non-canonical NF-κB signaling in response to MAC. ZFYVE21 induction required its sequestration on Rab5+MAC+ vesicles in a Rab5- and PI(3)P-dependent manner to promote vesicular pAkt recruitment, a requisite step for NIK stabilization[1].

**ZFYVE21 modulates PI(3,4,5)P3 by vesicular removal of PTEN**. Akt recruitment to activated membranes requires association of its pleckstrin homology domain with phosphatidylinositol 3,4,5 tris-phosphate (PI(3,4,5)P3)[15]. We hypothesized that ZFYVE21 promoted pAkt recruitment by modulating PI(3,4,5)P3 levels on MAC+Rab5+ vesicles. To test this, we used fluorescent lipid-binding reporter proteins to assess time-dependent alterations of PIs on MAC+Rab5+ endosomes using flow cytometry. HUVECs were co-transduced with a fluorescent Rab5 fusion protein, either Rab5-RFP or Rab5-GFP, and with a lipid-binding fusion protein, either 2XFYVE-GFP-binding PI(3)P or PH-RFP-binding PI(3,4,5)P3. The 2XFYVE-GFP and PH-RFP constructs contained PI(3)P and PI(3,4,5)P3 lipid-binding motifs derived from HRS[16–18] and Akt1[19–22], respectively, whose binding specificities have been heavily validated. Co-transduced HUVECs were then treated with vehicle or PRA sera containing C9-AF647. ECs were sonicated and mean fluorescent intensities (MFIs) of C9+Rab5+ events were analyzed by flow cytometry for association of 2XFYVE-GFP or PH-RFP (Supplementary Fig. 2a). Basally, MFIs of Rab5+FYVE+ were significantly higher than that of Rab5+PH+ (Supplementary Fig. 2b). Both C9+Rab5+2XFYVE+ and C9+Rab5+PH+ MFIs were significantly induced with PRA treatment at 30 min, a time point at which NIK was stabilized (Supplementary Fig. 2b). The association of 2XFYVE-GFP and PH-RFP with Rab5+ endosomes were attenuated in a dose-dependent manner upon PI(3)P and PI(3,4,5)P3 lipid depletion by KU-55933 and LY294002, respectively (Supplementary Fig. 2c). Both reporters showed moderate correlations when compared to PI-targeted Ab staining by confocal microscopy (Supplementary Fig. 2d).

We next performed a kinetic analysis of the PI content of C9+Rab5+ endosomes. We found increased PI(3)P:PI(3,4,5)P3 ratios at 5 min compared to pretreatment (Fig. 1q). This ratio subsequently became inverted by 15 min and persisted in this manner until 30 min, consistent with the interpretation that there is an early enrichment of PI(3)P on C9+Rab5+ vesicles that facilitated ZFYVE21 recruitment, followed by PI(3,4,5)P3 enrichment causing late recruitment of pAkt. Parallel confocal microscopic studies combined with the same reporters showed increased total PI(3)P and PI(3,4,5)P3 levels (Fig. 1r, right) on MAC+Rab5+ vesicles upon PRA treatment (Fig. 1r, left). With ZFVE21 siRNA, PI(3,4,5)P3 staining (Fig. 1r, left) and PH+ reporter association (Fig. 1r, right) with MAC+Rab5+ endosomes were reduced. These data indicated that ZFYVE21 increased relative levels of PI(3,4,5)P3 on MAC+Rab5+ endosomes at the expense of PI(3)P.

We sought to determine the mechanism by which ZFYVE21 increased vesicular PI(3,4,5)P3. A re-analysis of the proteome of MAC+Rab5+ vesicles revealed that PTEN, a PI(3,4,5)P3 phosphatase, showed decreased spectral counts in MAC+Rab5+ vs control Rab5+ endosomes (Fig. 2a). We hypothesized that MAC reduced vesicular PTEN stability to promote pAkt recruitment. To test this, we initially established a global role for PTEN for MAC-induced non-canonical NF-κB. We found that, in PRA-treated samples, PTEN siRNA not only potentiated the levels of pAkt, as expected, but also increased NIK (Fig. 2b, left) and NIK-dependent gene expression (Fig. 2b, right),

indicating that PTEN constrained pAkt-dependent non-canonical NF-κB activation. To define a role for endosome-associated PTEN, a population previously shown to modulate pAkt activity[23], we assessed PTEN levels on Rab5+ vesicles from HUVECs transduced with Rab5 DN or Rab5 WT. Basally, Rab5+ endosomes from Rab5 DN EC showed decreased vesicular PTEN vs Rab5 WT controls (Fig. 2c, left, lane 3 vs lane 1), a finding corroborated in parallel confocal microscopy studies (Fig. 2c, right). However, upon PRA treatment, Rab5 activity was required for concomitant loss of PTEN and induction of NIK (Fig. 2c, left, lane 2 vs lane 1). We confirmed that PRA reduced the stability of vesicular PTEN by means of co-IP studies (Fig. 2d), by confocal microscopy (Fig. 2e), and by enzymatic analyses of PTEN phosphatase activity, i.e., conversion of PI(3,4,5)P3 to PIP2, on Rab5+ endosomes (Fig. 2f). These data identified dual roles for the active form of Rab5, i.e., maintenance of PTEN on Rab5+ vesicles under basal conditions and removal of Rab5-associated PTEN via ZFYVE21-mediated signaling following complement activation.

**SMURF2-dependent degradation of vesicular PTEN**. We surmised that reduced PTEN stability on MAC+Rab5+ endosomes was mediated by a ubiquitin-mediated pathway. We observed time-dependent increase in K48 polyubiquitinylation of PTEN following PRA treatment (Fig. 2g). ECs transfected with a ubiquitin DN protein (Ub-DN) were unable to recruit pAkt and NIK, indicating that the process of ubiquitinylation was required (Fig. 2h). To determine whether vesicular or cytoplasmic pools of PTEN became ubiquitinylated, we overexpressed full-length vesicle-associated PTEN or a ΔPTEN mutant lacking the lipid-targeting C2 domain, a mutation preventing vesicular association of PTEN[23]. We observed that vesicle-associated PTEN underwent inducible polyubiquitinylation with complement treatment, whereas cytoplasmic PTEN did not (Fig. 2i). These data collectively demonstrated that vesicle-associated pools of PTEN underwent polyubiquitinylation, a process required for non-canonical NF-κB activation.

We next sought to identify the E3 ubiquitin ligase responsible for removing PTEN from the endosome. NEDD4 family members like NEDD4-1[24] and WWP2[25] were previously identified as E3 ubiquitin ligases for PTEN. We cross-referenced our prior siRNA screening hits[1] with NEDD4 family members, and we identified SMURF2 as a candidate. SMURF2 is implicated in transforming growth factor (TGF)-β signaling and maintenance of genomic stability via ubiquitinylation of the TGF-β receptor[26] and Ring Finger Protein 20 (RNF20)[27], respectively. To investigate a role for SMURF2 in ubiquitinylating PTEN, we initially tested the effects of SMURF2 knockdown on PTEN ubiquitinylation. We found that SMURF2 siRNA resulted in decreased polyubiquitinylation of immunoprecipitated PTEN (Fig. 3a). We subsequently assessed the role for SMURF2 on pAkt and NIK. HUVECs transfected with SMURF2 siRNA retained Rab5-associated PTEN to levels equivalent to vehicle-treated controls (Fig. 3b lane 3 vs lane 1). The retention of Rab5-associated PTEN in the presence of SMURF2 siRNA concomitantly blocked recruitment of pAkt and NIK to Rab5+ endosomes (Fig. 3b), leading to globally reduced levels of these proteins (Fig. 3c). SMURF2 DN or SMURF2 siRNA blocked pAkt and NIK (Fig. 3c) but not ZFYVE21 recruitment to Rab5 endosomes (Fig. 3d), whereas ZFYVE21 siRNA blocked SMURF2, pAkt, and NIK (Fig. 3d), indicating that SMURF2 was recruited to Rab5+ vesicles downstream of ZFYVE21 in a ZFYVE21-dependent manner. SMURF2, when co-incubated with recombinant FLAG-tagged ZFYVE21, did not show direct binding with ZFYVE21 (Fig. 3e), suggesting that SMURF2 recruitment required a yet undefined upstream intermediary. To

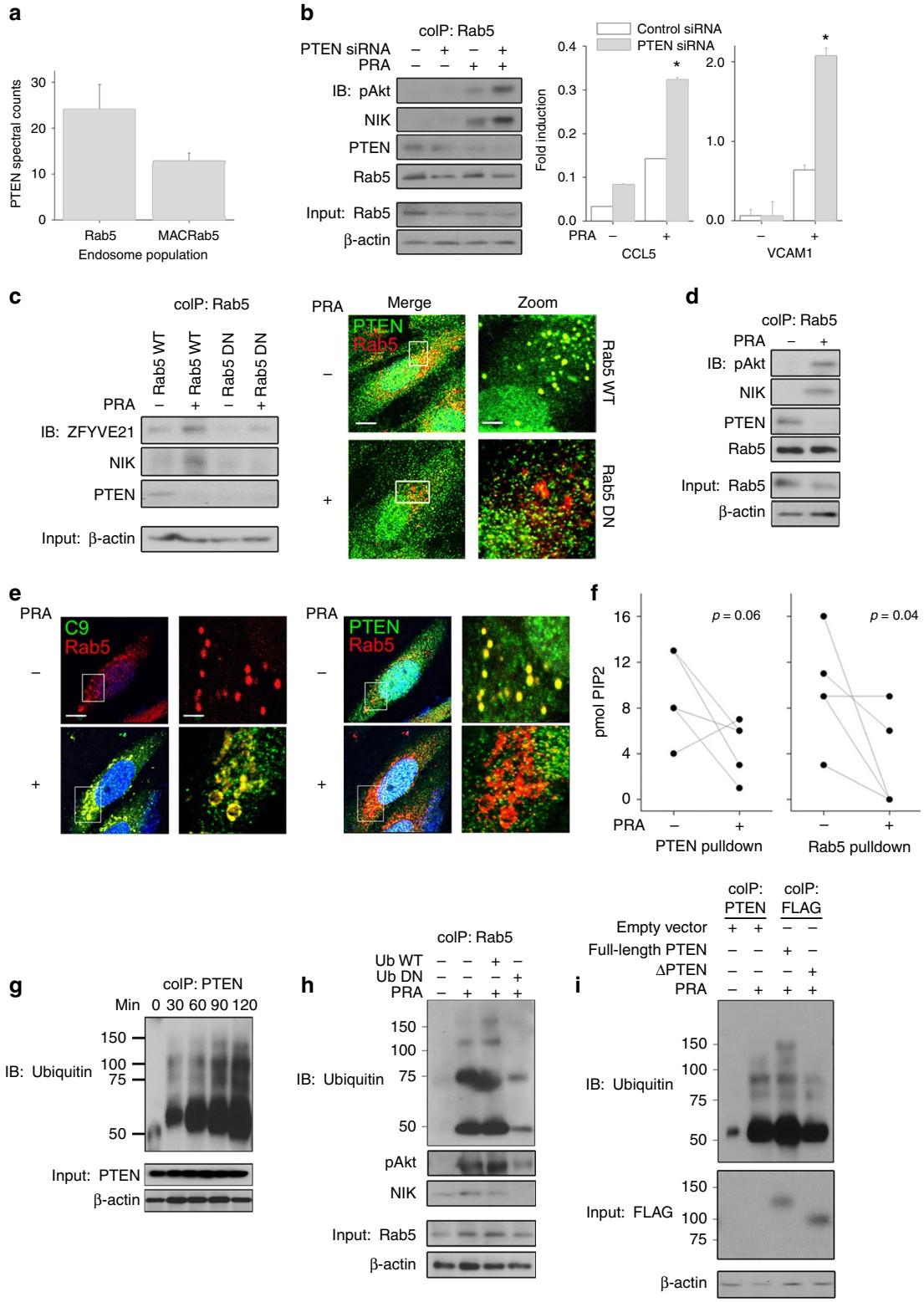

confirm that SMURF2 could ubiquitinylate PTEN, we performed in vitro ubiquitinylation assays. We identified UbcH5a, UbcH5b, UbcH5c, UbcH7, and UbcH8 as E2 ubiquitin ligases that could mediate this process (Fig. 3f). Using UbcH5c, we performed in vitro ubiquitinylation assays and found that SMURF2 WT but not SMURF2 DN or negative controls could catalyze PTEN ubiquitinylation (Fig. 3g). We observed that MG132 but

not chloroquine rescued PRA-induced degradation of PTEN (Fig. 3h), implicating proteasomes but not lysosomes as the predominant degradative mechanism for PTEN. Cumulatively, these data indicated that ZFYVE21-mediated recruitment of SMURF2 promotes proteasome-mediated degradation of vesicle-associated PTEN to alter endosomal PI composition to favor pAkt recruitment.

**Fig. 2** ZFYVE21 modulates PI(3,4,5)P3 by vesicular removal of PTEN. Spectral counts of PTEN in vesicular proteomes (**a**). Human umbilical vein endothelial cells (HUVECs) transfected with control or PTEN siRNA were treated with panel reactive antibody (PRA) sera for 30 min prior to Rab5 co-immunoprecipitation and western blotting (**b**, left) or with PRA sera for 4 h prior to quantitative reverse transcriptase PCR (qRT-PCR) analysis (**b**, right). HUVECs stably transduced with Rab5 WT or Rab5 DN constructs were treated with PRA sera for 30 min prior to Rab5 co-immunoprecipitation (**c**, left) or confocal microscopic staining (**c**, right, left scale bar: 10 μm, right scale bar: 342 nm). HUVECs were treated with PRA for 30 min prior to downstream analyses by Rab5 co-immunoprecipitation (**d**), confocal microscopic staining (**e**, left scale bar: 10 μm, right scale bar: 251 nm), and enzymatic assays assessing vesicular PTEN activity (**f**). Co-immunoprecipitation studies were performed as indicated (**g–i**). *$p < 0.05$, analysis of variance. Proteomic analyses were performed three times using three separate HUVEC donors. qRT-PCR analyses were performed using technical triplicates and was repeated two times. *$p < 0.05$, Student's t test. PTEN enzymatic activity assays utilized technical replicates and was repeated two times. Confocal microscopic analyses analyzed ≥5 cells per experiment, and each experiment was repeated three times. PTEN enzymatic assays utilized technical duplicates and was repeated using three HUVEC donors. All western blot and co-immunoprecipitation assays were conducted three to four times. Representative data shown

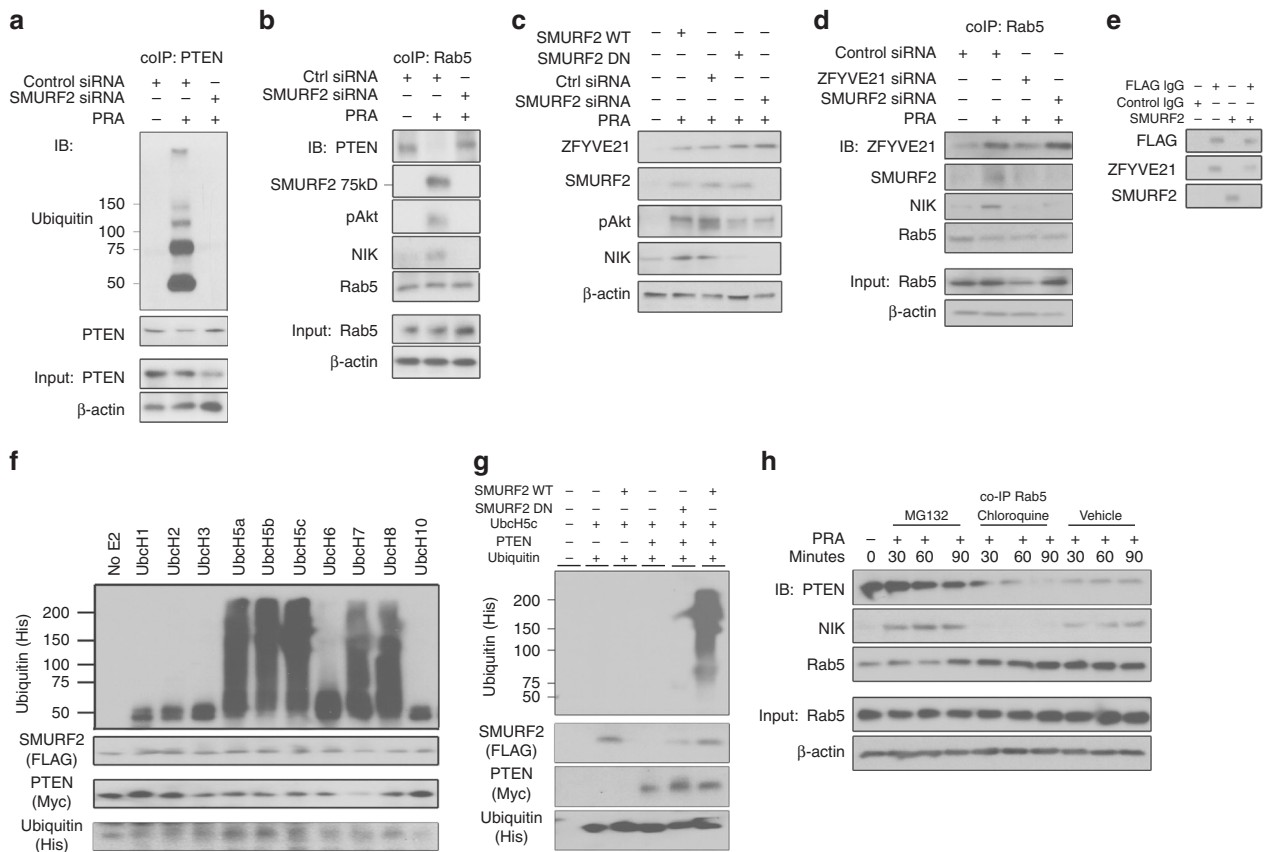

**Fig. 3** SMURF2-dependent degradation of vesicular PTEN. PTEN co-immunoprecipitations (co-IPs) were performed on human umbilical vein endothelial cells (HUVECs) treated for 30 min with panel reactive antibody (PRA) sera and blotted as indicated (**a**). Rab5 co-IPs were performed on PRA-treated HUVECs transfected with control or SMURF2 siRNA (**b**). Western blot analysis of whole-cell lysates of HUVECs transiently transfected with SMURF2 WT, SMURF2 DN, control siRNA, and SMURF2 siRNA (**c**). HUVECs transfected with control, SMURF2, or ZFYVE21 siRNA were treated with PRA and Rab5 co-IPs were performed (**d**). Recombinant FLAG-tagged ZFYVE21 (1 μg) and/or recombinant SMURF2 (1 μg) protein was incubated for 30 min at room temperature and analyzed by western blot (**e**). In vitro kinase assays were performed using various E2 ubiquitin ligases with SMURF2 protein (1 μg) as the E3 ubiquitin ligase and PTEN (1 μg) as substrate (**f**). In vitro kinase assays were performed using isolated SMURF2 WT or SMURF2 DN as E3 ubiquitin ligases, UbcH5c as the E2 ubiquitin ligase, and PTEN as substrate as indicated (**g**). HUVECs were pretreated with MG132 for 4 h or chloroquine for 30 min prior to PRA treatment at the indicated times prior to Rab5 co-IP and western blot analysis (**h**). Western blot analyses were repeated two to four times. Confocal microscopic studies analyzed ≥4 cells per group and was repeated three times. In vitro kinase assays were repeated twice. Representative data shown

**The Rab5-ZFYVE21-SMURF2 signaling axis occurs in vivo**. To recapitulate our findings in vivo, we utilized a previously described collagen–fibronectin gel model to form human microvessels in SCID/bg mice. HUVECs were suspended in collagen–fibronectin gels and implanted subcutaneously. Over a period of 2–4 weeks, embedded HUVECs spontaneously self-assemble into human EC-lined microvessels anastamozing with native murine vasculature[28]. We subsequently investigated the roles of Rab5, ZFYVE21, and SMURF2 in this model by implanting HUVECs transduced with Rab5 WT or Rab5 DN constructs, control or ZFYVE21 shRNA, or control or SMURF2 shRNA. Twenty-four hours prior to harvest, mice were intravenously injected with PRA sera. Upon harvest, microvessels from all groups showed staining for C4d, indicative of complement activation (Supplementary Fig. 3a, top). The PRA-treated Rab5 WT EC but not Rab5 DN EC upregulated ZFYVE21 (Supplementary Fig. 3a, bottom), HUVECs containing control but not ZFYVE21 shRNA upregulated SMURF2 (Supplementary

Fig. 3b), and HUVECs containing control but not SMURF2 shRNA upregulated NIK (Supplementary Fig. 3c). These data recapitulated the sequence of our in vitro findings, indicating that Rab5-ZFYVE21-SMURF2 signaling can occur in vivo.

**Pharmacologic depletion of PI(3)P blocks ZFYVE21 expression.** Using cultured human ECs, we then tested the functional effects of ZFYVE21 and SMURF2 on EC activation, a process occurring as a result of NIK stabilization[2]. ECs transfected with ZFVYE21 siRNA showed significantly decreased the expression of NIK- and non-canonical NF-κB-dependent pro-inflammatory genes (Fig. 4a) and showed decreased ability to ability to activate alloimmune CD4+ T cells in EC:T cell cocultures (Fig. 4b).

Posttranslational stabilization of ZVYE21 PI(3)P-dependent sequestration on MAC+Rab5+ endosomes (Fig. 1). These data pointed to a rational means to block ZFYVE21, i.e., via vesicular depletion of PI(3)P. We thus sought to test the premise that ZFYVE21 inhibition via PI(3)P lipid depletion could ameliorate EC activation in vitro and in vivo. To test this, we assessed SAR405 and PIKIII, selective inhibitors of class 3 PI3Ks that mediate conversion of PIP to PI(3)P[27], as well as KU-55933, which has been reported to not only deplete endogenous PI(3)P but also carries inhibitory effects on class II PI3Ks and ataxia telengiectasia mutated, a PI3K-like kinase[13]. At doses previously reported to selectively inhibit class 3 PI3K/Vps34[29], we found that PIKIII (5 nM) but not SAR405 (5 nM) significantly depleted PI(3)P lipid content on MAC+Rab5+ endosomes in HUVECs (Fig. 4c). KU-55933 (20 μM), like PIKIII, also significantly reduced PI(3)P lipid reporter activity (Fig. 4c). Concordant with these findings, we found that PI(3)P lipid depletion by PIKIII but not SAR405 showed a high degree of inhibitory activity on ZFYVE21 induction (Fig. 4d), inflammatory gene upregulation (Fig. 4e), and EC-mediated activation of alloimmune CD4+CD45RO+ T cells (Fig. 4f).

In an extension of these studies, we performed a bioinformatics search of Food and Drug Administration (FDA)-approved compounds with annotated effects on both PI metabolism[30] and Akt[31] and identified miltefosine (hexadecylphosphorylcholine), which is FDA approved for visceral leishmaniasis and cutaneous breast cancer metastasis. We found that, similar to PIKIII and KU-55933, miltefosine decreased 2XFYVE-GFP reporter MFIs on C9+Rab5+ endosomes (Fig. 4g). In contrast to another annotated Akt inhibitor, Akt VIII, miltefosine additionally blocked both recruitment of ZFYVE21 and SMURF2 to Rab5+ vesicles along with pAkt and NIK, whereas Akt VIII did not (Fig. 4h), indicating that, congruent with our bioinformatics search parameters, miltefosine displayed an ability to modulate both PI content (Fig. 4g) and Akt activation. Miltefosine selectively blocked NF-κB luciferase activity induced by PRA but not TNF-α, which activates canonical NF-κB through TNFR1 receptors (Fig. 4i), and significantly inhibited NIK-dependent inflammatory genes (Fig. 4j), EC-mediated activation of allogeneic memory CD4+ T cells (Fig. 4k), and adhesion of CD4+CD45RO+ T cells to PRA-treated EC monolayers under flow conditions (Fig. 4l). These data showed that PI(3)P lipid depletion may attenuate ZFYVE21-mediated EC activation and EC-mediated activation and recruitment of alloimmune CD4+ T cells. Between miltefosine and PIKIII, drugs showing the strongest effects in blocking ZFYVE21 and ZFYVE21-mediated EC activation, we chose miltefosine for further downstream and in vivo analyses due to its well-documented safety profile and its long half-life (>30 days). These features allowed a priori design of a non-toxic regimen that did not require multiple dosing, thereby limiting drug exposure solely to complement-treated human artery xenografts in a retransplantation model below.

**ZFVYE21 regulates complement-mediated inflammation in vivo.** Complement activation is involved in the pathogenesis of solid organ transplant rejection[32,33]. To assess functional effects of ZFYVE21-associated signaling in vivo, we utilized a humanized mouse model of allograft vasculopathy, a T cell-mediated process exacerbated by terminal complement activation on ECs[5]. In this model, human coronary arteries are implanted into the infrarenal aortae of pairs of immunodeficient SCID/bg mice. PRA is then injected into one of the mouse hosts and IgG-depleted PRA is administered to the other. Approximately 24 h later, these paired artery segments are retransplanted into immunodeficient mice engrafted with allogeneic lymphoid cells. Over 2 weeks, retransplanted human arteries develop T cell-mediated allograft vasculopathy that is exacerbated by PRA sera[2]. The retransplantation strategy prevents potential confounding interactions between PRA sera and human T cells or Fc receptor-bearing immune cells since PRA sera is not introduced into the second host mouse.

Initially, to assess ZFYVE21-associated signaling, mice bearing implanted arteries were treated with PRA or IgG-depleted PRA and then harvested for analysis 24 h later, i.e., without retransplantation. PRA-treated xenografts showed increased intimal C4d, a marker of complement activation, and ZFYVE21, SMURF2, and NIK (Fig. 5a). We found that miltefosine administration blocked the ability of PRA to upregulate intimal ZFYVE21 and E-selectin, an adhesion molecule upregulated by activated ECs (Fig. 5b). We then tested the consequences of perturbing ZFVYE21 signaling in vivo. Paired mice bearing human arterial xenografts were pretreated with either vehicle or miltefosine. Both animals were then treated with PRA sera for 24 h and then harvested and retransplanted into mouse hosts that were previously engrafted with adoptively transferred human peripheral blood mononuclear cells (PBMCs) from a donor allogeneic to the artery donor. In this experiment, both hosts received PRA and are expected to bear arteries activated by MAC, but only one of the animals received miltefosine prior to retransplantation. Owing to its >30 day half-life, following retransplantation, re-dosing of miltefosine was not required, thus limiting drug exposure solely to the retransplanted xenograft. Relative to vehicle controls, miltefosine-treated grafts showed significant reduction in intimal T cell infiltration (Fig. 5c). Further, miltefosine-treated grafts showed markedly reduced neo-intimal lesion area and concurrently increased luminal cross-sectional area (Fig. 5d). In the absence of PRA injection, miltefosine did not significantly alter neointimal CD3+ T cell infiltration (Supplementary Fig. 4a) or neointimal cross-sectional area (Supplementary Fig. 4b), indicating that basal treatment with miltefosine did not block inflammatory responses mediated by T cells reacting to unactivated ECs. Collectively, our data showed that pharmacologic inhibition of ZFYVE21 in ECs with miltefosine attenuated MAC-induced EC activation, EC-mediated recruitment of alloimmune T cells, and T cell-mediated allograft vasculopathy, a condition exacerbated by complement activation.

**Correlations of ZFYVE21 and complement in patient tissues.** NIK is a marker for non-canonical NF-κB activation[34]. However, NIK may be stabilized in ECs either by ligand:receptor interactions involving TNFRSF signaling[4] or by MAC (Fig. 1c)[2]. We tested the ability of ZFYVE21 to discriminate between these two routes of NIK stabilization in ECs treated with either LIGHT, a TNFRSF ligand binding LTβR, or with PRA sera to induce MAC assembly on EC. Our data showed that, while both LIGHT and PRA stabilized NIK, only PRA selectively upregulated ZFYVE21 (Fig. 6a). Based on these data, we hypothesized that ZFYVE21

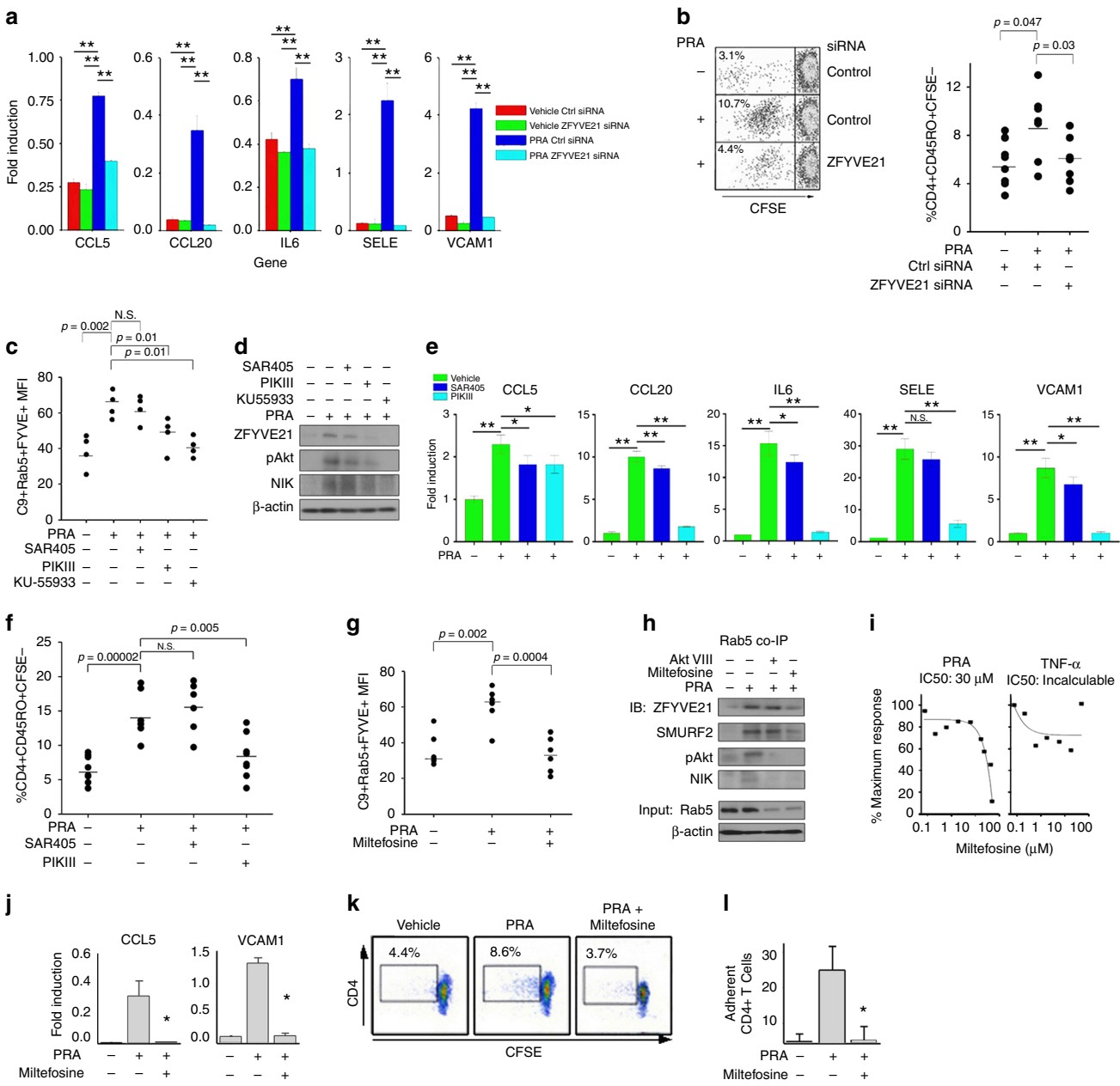

**Fig. 4** Pharmacologic depletion of PI(3)P blocks ZFYVE21 expression. Human umbilical vein endothelial cells (HUVECs) transfected with control or ZFYVE21 siRNA were treated with panel reactive antibody (PRA) for 4 h prior to quantitative reverse transcriptase PCR (qRT-PCR) (**a**) or prior to EC:T cell coculture for 7 days (**b**). *$p < 0.05$, analysis of variance. Lipid reporter mean fluorescent intensity (MFI) of PI(3)P (2XFYVE-GFP) on C9+Rab5+ vesicles following drug pretreatment with SAR405 (5 nM), PIKIII (5 nM), and KU-55933 (10 μM) for 30 min prior to addition of PRA sera for 30 min (**c**). Western blotting of HUVECs pretreated with inhibitors for 30 min prior to addition of PRA for 30 min (**d**). qRT-PCR analysis of HUVECs treated with PRA sera for 4 h following drug pretreatment (**e**). **$p < 0.001$. EC:T cell cocultures were performed using HUVECs pretreated with drug(s), and the percentage of proliferating memory T cells were calculated (**f**). Lipid reporter MFI of PI(3)P (2XFYVE-GFP) on C9+Rab5+ vesicles following drug pretreatment with miltefosine (25 μM) for 30 min prior to addition of PRA sera for 30 min (**g**). Western blotting of HUVECs pretreated with miltefosine or AktIII for 30 min prior to addition of PRA for 30 min (**h**). HUVECs stably transduced with nuclear factor (NF)-κB luciferase reporter were treated with varying doses of miltefosine prior to addition of PRA (25% v/v in gelatin veronal buffer) or tumor necrosis factor-α (10 ng/mL) for 6 h (**i**). qRT-PCR of HUVECs following PRA sera treatment for 4 h with miltefosine (25 μM, **j**). EC:T cell cocultures using HUVECs pretreated with miltefosine prior to addition of CFSE-labeled CD4+CD45RO+ T cells (**k**). CD4+CD45RO+ T cells were injected over confluent PRA-treated HUVEC monolayers and adherent cells were quantified (**l**). qRT-PCR analyses were performed using technical triplicates, and each experiment was repeated 2–4 times. EC:T cell cocultures used 6 technical replicates for each group and each experiment was repeated 2–3 times. Lipid reporter analyses used 4–6 technical replicates and each experiment was repeated 3 times. Western blot analyses were repeated two to five times. NF-κB luciferase assays were performed using technical triplicates, and each experiment was repeated three times. Flow chamber studies were performed using technical triplicates with quantifications of ≥3 high-powered fields per sample, and each experiment was performed twice. Representative data shown **$p < 0.001$

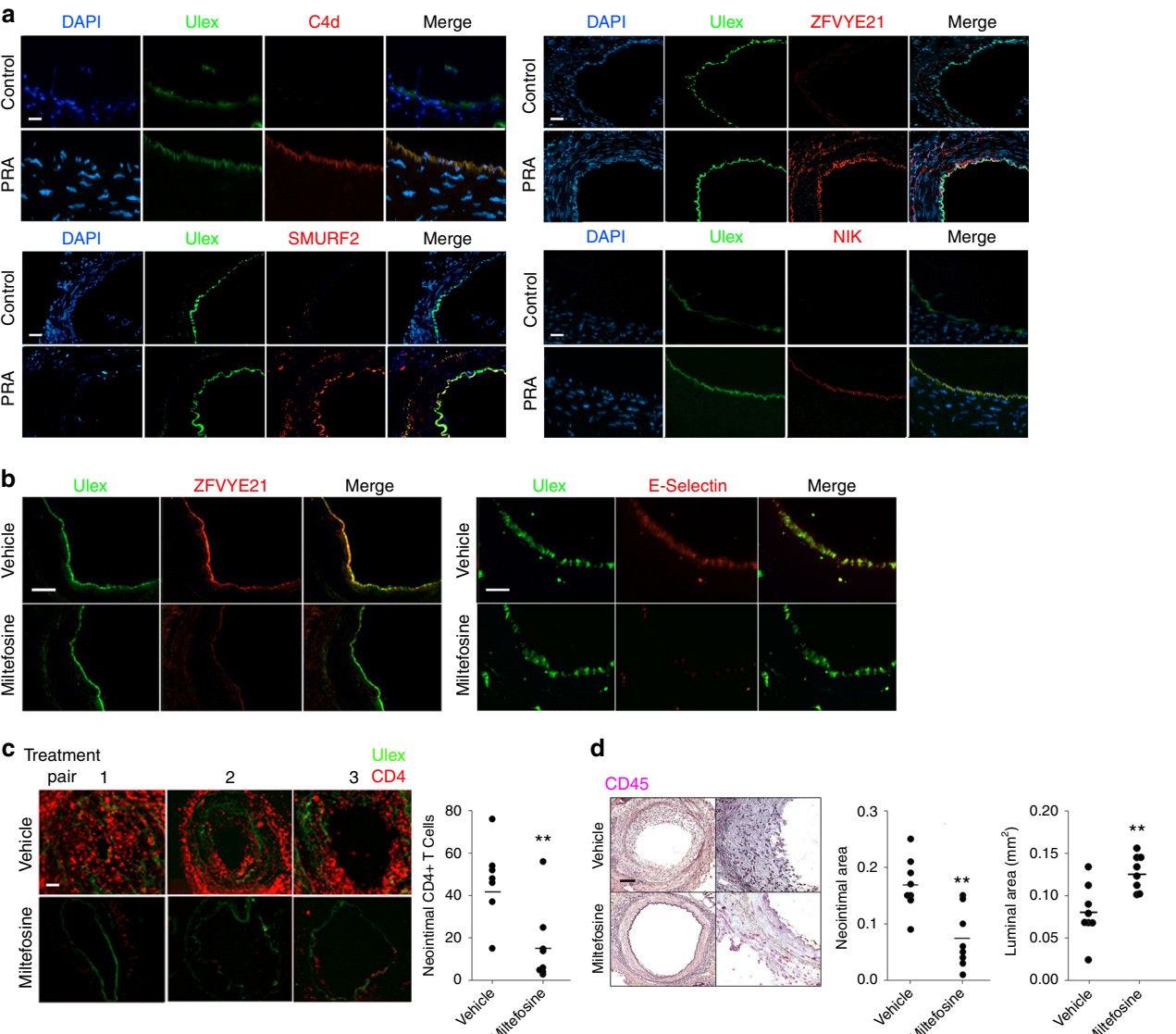

**Fig. 5** ZFVYE21 regulates complement-mediated inflammation in vivo. Coronary artery segments were implanted as interposition grafts in descending aortae of SCID/bg immunodeficient mice who received 200 μL IgG-depleted (*n* = 4) or intact panel reactive antibody (PRA) sera (*n* = 4) intravenous prior to graft harvest and immunofluorescence analysis as indicated (**a**, scale bar: left 200 μm, right 400 μm). Human coronary artery xenografts were pretreated with miltefosine (40 mg/kg once daily intraperitoneally for 3 days) prior to injection of PRA sera. Grafts were harvested 24 h afterwards and analyzed for intimal expression of ZFYVE21 and E-selectin (**b**, *n* = 3, scale bar: left 200 μm, right 400 μm). Human coronary artery segments treated with vehicle (*n* = 8) or miltefosine (*n* = 8) were exposed to PRA sera and then reparked as interposition xenografts in the descending aortae of SCID/bg immunodeficient mice engrafted with human T cells. Fourteen days posttransplant, neointimal CD4+ T cells were enumerated (**c**, scale bar: 800 μm), and neointimal and luminal areas were calculated (**d**, scale bar: 800 μm). **p < 0.001, Student's *t* test **p < 0.001

expression could specifically identify complement-bound ECs in patient tissues.

To test this hypothesis, we examined biopsies from conditions associated with complement activation. We first analyzed renal biopsies from patients with antibody-mediated rejection (ABMR) where complement activation is known to occur. Concordant with findings in the Human Protein Atlas (www.proteinatlas.org), we found bright ZFYVE21 staining in glomeruli in controls that was confined almost exclusively to Ulex− cells and no ZFYVE21 staining was observed in Ulex+ cells in peritubular capillaries. In contrast, ZFYVE21 was strongly upregulated in Ulex+C4d+ peritubular capillary ECs in ABMR biopsies (Fig. 6b). We next examined synovium from patients with either high- or low-titer anti-rheumatoid factor antibody rheumatoid arthritis (RA) or osteoarthritis (OA) for NIK, ZFYVE21, and

MAC (polyC9). Both forms of RA are inflammatory diseases associated with high levels of infiltrating leukocytes and complement activation. OA typically shows much less synovial inflammation but is associated with complement activation[35]. All RA and OA specimens showed EC deposition of MAC and ZFVYE21 (Fig. 6c), which was lacking in relevant controls. Spatial correlations of ZFYVE21 (Fig. 6d) and NIK (Fig. 6e) with ECs were quantified (Fig. 6f), revealing that ZFYVE21 showed significantly higher endothelial colocalization vs NIK in all three patient cohorts. ZFYVE21 had more highly restricted colocalization with MAC than NIK, especially in low-titer RF RA patients and OA (Fig. 6g). Upon observer-blinded analyses, high- and low-titer RF RA samples contained significantly more CD45+ immune cell infiltration (Fig. 6h). Consistent with the ability of various TNFSRSF ligand:receptor interactions to cause NIK

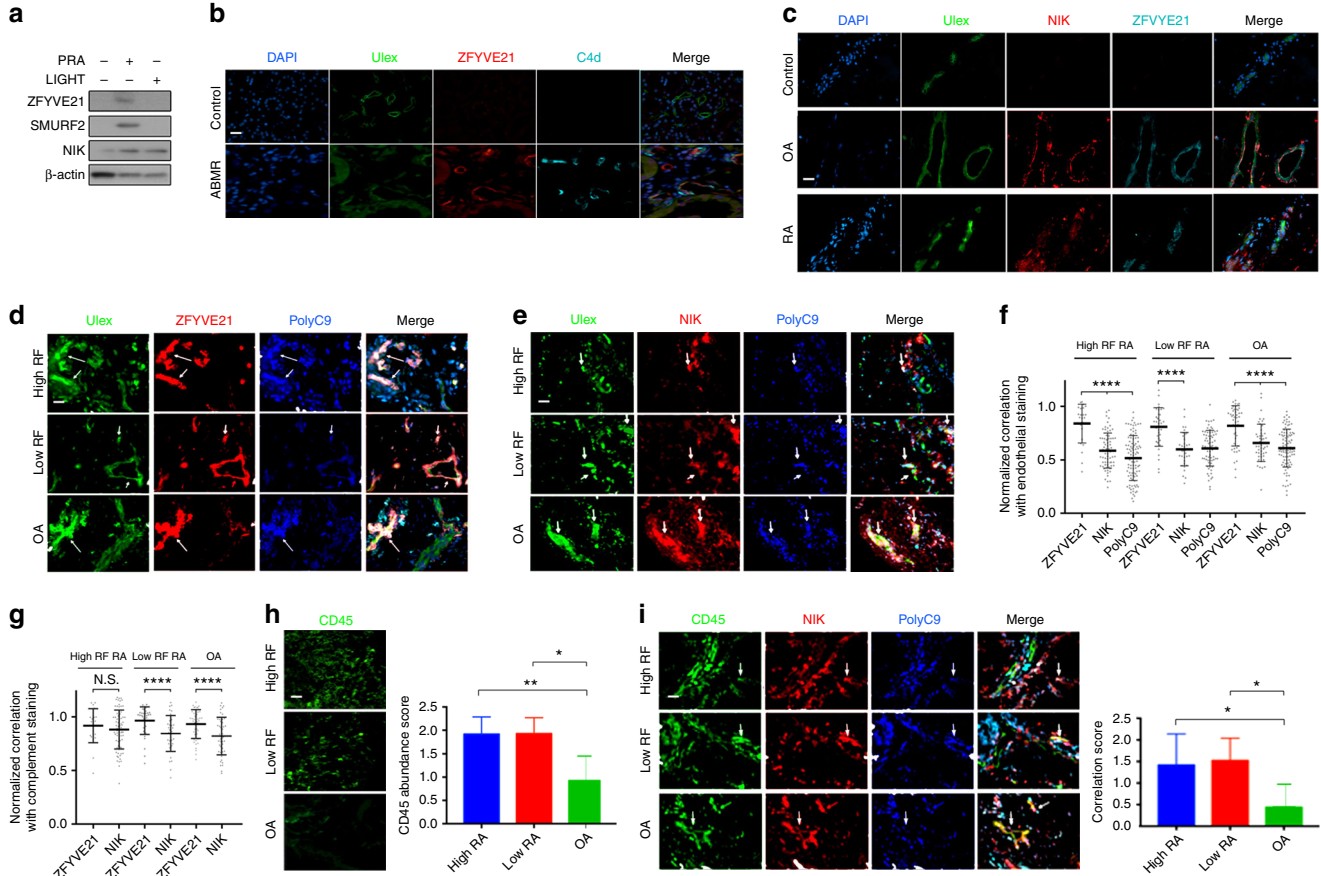

**Fig. 6** Correlations of ZFYVE21 and complement in patient tissues. Western blots of human umbilical vein endothelial cells treated with either panel reactive antibody (30 min, 25% v/v in gelatin veronal buffer) or LIGHT (12 h, 50 ng/mL in EGM2 media, **a**). Kidney biopsies from renal transplant patients (n = 12) were stained as indicated (**b**). Synovial biopsies from osteoarthritis (n = 11), low-titer rheumatoid factor antibody rheumatoid arthritis (RF RA; n = 8), and high-titer RF RA (n = 8) were stained as indicated (**c–e**) and spatial correlations between ZFVYE21 or nuclear factor-inducing kinase (NIK) with endothelial cells (Ulex, **f**) and membrane attack complex (polyC9, **g**) were assessed using the computer software (scale bar: 200 μm). ****$p < 1 \times 10^{-5}$, Student's $t$ test. CD45 abundance was assessed (**h**) and correlated with NIK staining (**i**) via blinded scoring from three independent observers. *$p < 0.05$, **$p < 0.001$. Western blotting was repeated three times. Analysis of variance statistical analyses were performed. Representative data shown *$p < 0.05$, **$p < 0.001$

stabilization independent of MAC, we observed significantly greater correlative staining between CD45 and NIK (Fig. 6i) when compared to OA specimens. Thus, in our analysis, the comparatively restricted staining of ZFYVE21 to complement-activated EC vs NIK may be explained by the heterogeneous staining of NIK in both EC and immune cells. These data indicated that, ZFYVE21 is a functional marker for complement activation in EC and may represent a more specific staining marker than NIK in certain complement-mediated disease conditions.

## Discussion

The present work and our cumulative studies advance the notion that, in an allogeneic (and not xenogeneic) system containing human MAC deposition on human EC, MAC is non-cytolytic but instead activates a pro-inflammatory signaling response[1,2,5], conceptually placing MAC as a de novo receptor. The present studies additionally demonstrate that ZFYVE21 is an inducible Rab5 effector that regulates EC activation, highlighting a previously undescribed role for Rab5 activity in promoting inflammation. These findings expand the functions of Rab5 effectors whose roles have been exclusively defined within homeostatic settings involving processes related to vesicular trafficking

and remodeling. To perform this dual function, Rab5 likely utilizes differential subsets of effectors to promote either homeostatic vs inflammatory processes. The mechanism(s) by which Rab5 selects effectors for downstream signaling remains unclear but may be related to (1) sequestration of relevant effectors to the appropriate compartment, e.g., recruitment to relevant endosome signaling compartment(s), and/or (2) inducible alterations in the repertoire of effectors available for binding Rab5, e.g., posttranslational upregulation of effector protein(s). The subset of Rab5 effectors promoting inflammation like ZFYVE21 thus represents molecular target(s) to prevent tissue injury.

Human coronary vessels affected by vasculopathy show antibody and complement deposition on the endothelium[36] and contain T cells but not B cells in a subendothelial position[37,38], suggesting an element of direct allorecognition. Our humanized mouse model not only faithfully recapitulates endothelial retention of MHC class II molecules but also biases toward direct allorecognition because "host" human myeloid cells are not effectively engrafted by adoptive transfer of PBMCs. With consideration to the above, we have previously used our current humanized mouse model to show that (1) nanoparticle-mediated inhibition of MHC class II on donor endothelium results in immunologic ignorance[39] and (2) PRA sera alone did not result

in allograft vasculopathy in the absence of T cells[2]. These data support the contention that allograft vasculopathy is a T cell-mediated disease involving direct allorecognition of graft ECs.

## Methods

**Cell culture, reagents, and culture treatments**. All protocols involving human biospecimens were approved by the Yale IRB and are compliant with all relevant ethical regulations. Informed consent was not required for any specimens as they were provided in a de-identified manner. HUVECs were obtained as tissue discarded material from the Department of Obstetrics and Gynecology at Yale New Haven Hospital. HUVEC culture, treatment with PRA, Western blot, real-time PCR, EC:T cell cocultures, flow chamber assays, and immunofluorescent staining were performed[1,2]. HUVECs were pretreated with Dynasore (80 μM Sigma), Pitstop2 (30 μM, Sigma), SAR405 (5 nM, Cayman Chemical), PIKIII (5 nM, Cayman Chemical), KU-55933 (Sigma), Akt VIII (5 μM, Cayman), miltefosine (Sigma), and chloroquine (Sigma, 30 mM) for 30 min prior to PRA sera treatment in in vitro experiments as indicated in the text. MG132 (25 μM, Selleck Chemicals) was added to EGM2 media for the indicated times.

For EC:T cell cocultures, HUVECs were grown in U-bottom 96-well microtiter plates, pretreated with interferon (IFN)-γ (50 ng/mL, Invitrogen), and subjected to drug or siRNA treatments as indicated in the text. On the day of the experiment, HUVECs were treated with or without PRA sera (25% v/v in gelatin veronal buffer) for 6 h prior to addition of isolated CD4+CD45RO+ T cells stained with CFSE (Molecular Probes) as described[2] at $1 \times 10^6$ cells/well at a volume of 200 μL in RPMI (Gibco) supplemented with 5% fetal bovine serum (FBS), 1.5% L-glutamine, and 1% penicillin/streptomycin. T cells were harvested 7–10 days after co-culture and analyzed by flow cytometry (FACSCalibur). For flow chamber studies[2], confluent HUVEC monolayers grown on glass coverslips were treated with PRA sera (25% v/v in gelatin veronal buffer) prior to washing and placement in EGM2 media overnight prior to the start of the assay. The following day, isolated CD4 +CD45RO+ T cells were flowed over confluent PRA-treated HUVEC monolayers as described[2] and adherent cells were quantified by immunofluorescence (I.F.).

PIP-coated beads were purchased commercially (Echelon Biosciences) and incubated with 1 μg ZFYVE21 protein (Origene) as per the manufacturer's recommendations. To determine ZFYVE21 binding to active Rab5, 20 μL protein A/G beads (ThermoFisher) were incubated with 10 μL Rab5 antibody at 4 °C overnight. Rab5 protein (1 μg, abcam) or protein lysates (10 μg) were incubated with antibody-conjugated protein A/G beads 4 °C overnight. Beads were washed and incubated with non-hydrolyzable GTPγS (100 μM) or GDP (100 μM, Sigma) for 30 min at room temperature prior to addition of ZFYVE21 (1 μg) for 30 min at room temperature[40]. To determine ZFYVE21 binding to SMURF2, 20 μL protein A/G beads (ThermoFisher) were incubated with 10 μL ZFYVE21 antibody (Atlas) at 4 °C overnight. ZFYVE21 (1 μg, Origene) protein was incubated with antibody-conjugated protein A/G beads 4 °C overnight. Beads were washed and incubated with SMURF2 (1 μg, Novus) for 30 min at room temperature. All incubations above were performed at a volume of 32 μL using RIPA buffer (Cell Signaling). Following the incubations, Laemli's buffer (12 μL) and 1 mM dithiothreitol (6 μL) were then added to samples, heated for 95 °C for 13 min, and subjected to western blotting. Antibodies used for western blotting were all used at 1:1000 dilution and included ZFYVE21 (Novus Biologicals, #H00079038-B01P), SMURF2 (Cell Signaling Technology, #12024), NIK (Cell Signaling, #4994), PTEN (Cell Signaling, #9188), pAktThr308 (Cell Signaling, #13038), active Rab5 (NewEast Biosciences, #26911), FLAG (Cell Signaling, #14793), Rab5 (Santa Cruz Biotechnology, #sc-46692), K46-conjugated ubiquitin (Cell Signaling, #3936), and β-actin (Sigma, #A5316-100 μL). Uncropped western blot films are shown in Supplementary Figs. 5–7, and corresponding densitometry analyses are shown in Supplementary Figs. 8–11. In vitro PTEN activity assays were performed using a commercially available kit according to the manufacturer's recommendations (Echelon). In vitro ubiquitinylation studies were performed with SMURF2 (1 μg) and PTEN (1 μg, Cayman) proteins using a commercially available kit (Ubiquitinylation Kit, Enzo).

For immunostaining, HUVECs were grown on glass coverslips, fixed, and permeabilized with ice-cold methanol for 15 min, blocked with PBS containing 0.1% Tween and 5% FBS for 1 h at room temperature (PBST), and stained with relB (Santa Cruz, #sc-226), PI(3)P (Echelon, #Z-P003), PI(3,4,5)P3 (Echelon, #Z-P345b), Sc5b-9 (Dako, M0777), PTEN (Cell Signaling #9188), NIK (Cell Signaling, #4994), and ZFYVE21 (Atlas, HPA055721) at 1:200 dilution for 4 °C overnight. Fluorescent-tagged secondary Abs (Molecular Probes) were used at 1:500 dilution for 1 h at room temperature in PBST.

**Animal studies**. All protocols were approved by the Yale Institutional Animal Care and Use Committee and are compliant with all relevant ethical regulations. Human coronary arteries were interposed into the descending aortae of adult female C.B-17 SCID/beige mice (Taconic, Hudson, NY) as described[2]. For drug treatment studies, mice received miltefosine (60 mg/kg) daily for 3 days prior to intravenous injection of 200 μL of neat PRA sera. Twenty-four hours after PRA injection, arteries were retransplanted into SCID/bg mice that had been previously injected with human PBMCs as described[2]. Fourteen days after retransplantation, grafts were harvested and analyzed.

**Viral transduction of EC**. A lentiviral NF-κB luciferase reporter (SA Biosciences, Valencia, CA) and lentivirus-encoded control shRNA, ZFYVE21 shRNA, and SMURF2 shRNA (Sigma) was transduced using two cycles of transduction at a multiplicity of infection of 10 in 80–90% confluent HUVEC cultures for 8 h followed by washing, culturing with EGM2-MV medium overnight. Knockdown efficiencies were confirmed by western blot prior to use in collagen gels in vivo.

**siRNA transfection of EC**. HUVECs were pretreated with IFN-γ for 48 h prior to siRNA transfection. siRNA (Dharmacon, Waltham, MA) targeting ZFYVE21, SMURF2, PTEN, or non-targeting siRNA (target sequence UAA CGA CGC GAC GAC GUA A) were purchased commercially (Dharmacon) and transfected into HUVECs at ~60–70% confluency in 24-well plates (BD Falcon). siRNAs were diluted at 40 nM concentration in Opti-Mem culture media (Gibco) and mixed at equal volume with RNAiMax transfection reagent (Invitrogen) diluted 1:50 in Opti-Mem for 45 min at room temperature. This mixture was then added to HUVEC cultures at 37 °C for 6 h prior to washing and buffer exchange with EGM2-MV. Cells were then analyzed by western blot, luciferase assay, qRT-PCR, or T cell functional assays 72 h after transfection.

**Quantitative RT-PCR**. RNA was isolated from treated HUVECs according to the manufacturer's specifications (Qiagen) and reverse transcribed (Applied Biosystems). Respective cDNA was amplified in a CFX Realtime System (Biorad) at a volume of 20 μL containing dilutions of 1:20 Taqman probe (Applied Biosystems), 1:2 Taqman Gene Expression Master Mix (Applied Biosystems), and 1:10 cDNA in ddH₂O. RT-PCR gene probes were purchased from Applied Biosystems and included: CCL5 (#Hs00174575_m1), CCL20 (#Hs01011368_m1), IL6 (#Hs00985639_m1), SELE (#Hs00950401_m1), VCAM-1 (#Hs01003372_m1), and GAPDH (Hs02758991_g1). For amplification, samples were heated to 50 °C for 2 min for once cycle, 95 °C for 10 min for one cycle, and then 40 cycles where samples were heated to 95 °C for 15 s proceeded by 60 °C for 1 min. Cycle times were subsequently normalized to the average GAPDH cycle time for all samples such that a relative decrease in cycle number by $x$ indicates gene induction by $2^x$.

**C9 fluorescent labeling and endosome sorting**. C9 protein purified from human serum (Sigma) was labeled with AlexaFluor647 (Invitrogen, #A20173) according to the manufacturer's specifications. A quantity of 800 μg of C9 protein was used per labeling reaction. Labeled C9 eluted from the manufacturer's column was then serially concentrated and re-diluted in PBS using five 30 min spins at $2100 \times g$ in Amicon Ultracel 10 K Centrifugal Filter Devices (EMD Millipore, Billerica, MA) and brought up in a final volume of 200 μL PBS. Prior to use, the amount of labeled C9 was calculated using the provided equations in the manufacturer's instructions (Molecular Probes, cat #A20173), showing that the moles of AF647 dye incorporated per mole C9 protein ranged between 0.52 and 1.21. Seventy-five microliters of AlexaFluor 647-labeled C9 was added to 500 μL PRA and 1925 μL gelatin veronal buffer and this mixture was added to IFN-γ pretreated HUVECs carrying a Rab5-GFP reporter or to HUVECs carrying fluorescent lipid reporters (PH-RFP or 2XFYVE-GFP) as below for 30 min. Twenty confluent T175 flasks treated as above were used for proteomic analyses. For lipid reporter experiments, individual C6 wells were used. Cells were harvested, washed, and resuspended in 0.5 mL endosome buffer [10 mM HEPES-NaOH pH 7.4, 1 mM EDTA, 0.25 M sucrose containing 1 protease inhibitor tablet (Roche) per 10 mL buffer] and mechanically disrupted by three freeze–thaw cycles followed by three cycles of sonication for 20 s. Subcellular fractions were identified and gated by forward scatter using a linear axis scale, and Rab5-GFP+C9-AlexaFluor 647+ subcellular fractions representing MAC+Rab5+ vesicles were sorted (FACSAria, Becton Dickinson) using the gating strategies indicated. Sorted vesicles were concentrated to a volume of 500 μL using Amicon filter centrifugation (EMD Millipore) using a 5 kDal cut-off prior to proteomic analysis.

For analysis of protein aggregates, two batches of C9 AF647 were tested. Seventy-five microliters of C9 AF647 were added to 2 mL gelatin veronal buffer to reproduce the concentration used to treat HUVECs for confocal microscopy analyses as described above. A portion of this supernatant, referred to in the text as C9 AF647 Pre, was added at 100 μL to HUVECs plated on glass coverslips in` C24 well plates for 30 min at 37 °C prior to staining for Rab5. C9 AF647 products were ultracentrifuged at $100,000 \times g$ for 1 h at 4 °C using a TLS55 rotor in an Optima TLX Ultracentrifuge (Beckman). Pellets were resuspended in 2 mL gelatin veronal buffer (C9 AF647 Post) and used to treat HUVECs as above. Absorbance was measured at 600 nm using a SpectraMax iD3 spectrophotometer (Molecular Devices).

**Reporter constructs**. The PH-RFP and 2XFYVE-GFP reporter constructs were a generous gift from Dr. Pietro de Camilli (Yale University School of Medicine, New Haven, CT). The PH-RFP and 2XFYVE-GFP constructs contain a PH domain derived from Akt1 showing preferential binding to PI(3,4,5)P3 > PI(3,4)P2 and a FYVE domain from HRS binding PI(3)P, respectively. pcDNA3-FLAG PTEN was a gift from Jaewhan Song (Addgene plasmid #78777). mCherry-Rab5WT was a gift from Gia Voeltz (Addgene plasmid #49201). mCherry-Rab5DN (S34N) was a gift from Sergio Grinstein (Addgene plasmid #35139). Rab5-GFP lentiviral particles were purchased commercially (AMS Biosciences). mCherry-FKBP-MTM1 was a

gift from Tamas Balla (Addgene plasmid #51614). pCMV5B-Flag-SMURF2 C716A was a gift from Jeff Wrana (Addgene plasmid #11747). pRK-Myc-SMURF2 was a gift from Ying Zhang (Addgene plasmid #13678). Ubiquitin WT was a gift from Rachel Klevit (Addgene plasmid #12647). pRK5-HA-Ubiquitin-K48R was a gift from Ted Dawson (Addgene plasmid #17604).

**Collagen–fibronectin gel model**. HUVECs were grown to confluency in T75 flasks, harvested, pelleted, and resuspended in 500 μL of a solution mixture containing 50 μL 10× M199 (Sigma), 50 μL of 1 mg/mL fibronectin (Millipore), 12.5 μL 1 M HEPES, 188 μL of 10 mg/mL NaHCO₃, 300 μL of 3.66 mg/mL Collagen (Corning), and 6 μL 1 M NaOH. Gels were solidified at 37 °C for 1 h prior to subcutaneous implantation into the flank of SCID/bg mice. Four weeks later, gels were harvested, flash frozen in OCT, sectioned, stained, and analyzed by I.F. as indicated. Staining Abs were used at 1:200 dilution including ZFYVE21 (Atlas), SMURF2 (Cell Signaling), NIK (Cell Signaling), Ulex (Vector Labs), and Sc5b-9 (Dako).

**Confocal microscopy and STED imaging**. Confocal microscopy studies were performed on an TCS SP8 confocal microscope (Leica). STED images were acquired using a commercial Leica TCS STED SP8 3X instrument. 594 and 650 nm laser lines were used to excite the respective dyes while a 775 nm laser line was used for depletion. Primary ZFYVE21 (1:200, Atlas Antibodies) and Rab5 (1:200, Santa Cruz) antibodies were used. Scanning was performed in a frame sequential manner in which the Atto647 secondary dye (1:1000, Sigma) was imaged first in confocal, then in STED, followed by imaging the Atto594 dye (1:1000, Sigma) in confocal, then in STED. The STED images from both color channels were analyzed using a wavelength-based algorithm[41] to detect the respective signals from each wavelength. The center positions of the spots were determined by a weighted centroid method, and the distribution histogram of the nearest distance between the spots in the two-color channels was calculated and reported.

**Immunofluorescence analysis**. Immunohistochemical staining of human coronary artery segments was performed[2]. Formalin-fixed paraffin-embedded renal biopsy sections ($n = 13$) were obtained as archival samples from the Yale Department of Pathology and did not require informed consent as the samples were provided in a de-identified fashion. Following deparaffinization and hydration, antigen retrieval was performed at 95 °C for 1 h in Antigen Unmasking Solution (VectorLabs) and stained using antibodies against ZFYVE21 (Atlas), NIK (Cell Signaling), Ulex (VectorLabs), and C4d (Novus) at 1:200 dilution at 4 °C overnight prior to addition of secondary Abs (1:500 as above for 1 h at room temperature. Healthy synovial tissues ($n = 3$, Articular Engineering) were used as staining controls. Frozen sections of human synovial biopsies were provided from archived tissue at the Brigham and Woman's Hospital, Boston, MA. Samples were divided into three groups: RA patients who expressed high titers of rheumatoid and anti-citrulline peptide autoantibody (High RA, $n = 8$), RA patients who expressed low levels of rheumatoid and anti-citrulline autoantibody (Low RA, $n = 8$), and OA ($n = 11$). Frozen sections were analyzed by immunofluorescent microscopy. Briefly, 5 μm sections were fixed in ice-cold acetone for 10 min, air-dried, washed 3 × 5 min in PBS, and then blocked in Ab Blocking Solution (BD Biosciences) for 1 h at room temperature. Primary antibodies were incubated overnight at 4 °C at 1:200 dilution with the following antibodies: NIK (abcam #7204), PolyC9 (Quidel #A239), and ZFYVE21 (Atlas #HPA055721, Ulex (Vector Labs, #B-1065). The next morning, slides were washed three times in PBS and incubated 1 h at room temperature with Dylight anti-mouse Ig 594 nm and Dylight anti-rabbit Ig 674 nm antibodies (Vector Labs). Following staining, slides were washed, air dried, and cover slipped using a DAPI mounting media (Immuno Gold with DAPI, Invitrogen). I.F. was visualized using a Zeiss Axiovery 200 M fluorescence microscope.

Image co-localization was determined in Matlab using a 2D correlation coefficient ("corr2"). Images were initially thresholded to remove background autofluorescence with the threshold determined by imaging tissue samples in the absence of fluorescent staining. The filtered images were then analyzed to determine the degree of spatial correlation between various image channels corresponding to the proteins of interest. For each patient sample, a minimum of four independent images were analyzed. Correlations were normalized against negatively and positively stained control samples to define correlation values of "0" and "1," respectively. In certain analyses, specimens were scored by three blinded investigators. Scoring for CD45: 0 = no infiltration, 1 = mild perivascular infiltration, 2 = moderate perivascular infiltration, and 3 = diffuse infiltration. Scoring for correlative CD45 and NIK staining: 0 = no correlation, 1 = <25% correlation, 2 = 50–75% correlation, 3 = >75% correlation.

**Statistical methods**. Statistical analyses were performed using the computer software (Origin, Northampton, MA). Absolute numbers and percentages of vesicles were analyzed by Student's $t$ test and Chi-squared analyses, respectively. $p$ values < 0.05 were considered statistically significant. Multiple comparison analyses were performed using analysis of variance. Standard deviations are reported throughout the text. Colocalization for confocal analyses was quantified using the ImageJ image analysis software (Bethesda, MD) with the Just Another Colocalisation Plugin (JACoP). Intermolecule distances were calculated using software developed in-house per the laboratory of Dr. Joerg Bewersdorf.

**Reporting summary**. Further information on research design is available in the Nature Research Reporting Summary linked to this article.

## Data availability
All data are available from the authors upon reasonable request. The proteomics datasets have been publicly deposited in the PRIDE Archive (Accession #: PXD013381).

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

## Acknowledgements

D.J.-w. and J.S.P. were both supported by a grant from AbbVie Pharmaceutical Research & Development (YAP-005-2013). Additional support was provided by grants from the NIH to J.S.P., T.D.M. (R01-HL051014), P.A.N. (P30 AR070253 and R01AR065538), and D.J.-w. (R01HL141137-01, R00HL125895, UL1TR001863, P30AR053495-01A1). We thank Dr. Verena Broeckner for collection, identification, and diagnosis of human renal biopsy sections for research at Addenbrooke's Hospital, Cambridge, UK with support from the NIHR Cambridge Biomedical Research Centre. We additionally thank the Center for Cellular and Molecular Imaging, Mark Lessard, and Joerg Bewersdorf for assistance in performing STED imaging. Mark Lessard and Joerg Bewersdorf were supported in part by NIH grant S10 OD020142.

## Author contributions

C.F. and L.L. performed western blot experiments. T.D.M. performed T cell adhesion experiments and Rab5 cloning experiments. K.L., W.F., R.L., and C.X. performed qRT-PCR studies. L.Q., G.L., G.T., and J.S.P. were involved in human artery retransplantation studies. Z.T. performed FACS-assisted endosome sorting studies. G.T.T., M.P., N.C.K.-S., J.M., and P.A.N. were involved in human biopsy analysis. D.J.-w. wrote the manuscript and designed all experiments.

## Additional information

**Competing interests:** The authors declare no competing interests.

