## [Peer Review File · Nature Communications]

Reviewers' comments:

Reviewer #1 - expert in allograft vasculopathy (Remarks to the Author):

This manuscript extends the authors' previous publications that have explored the contribution of complement and alloantibody to allograft vasculopathy.

In their previous work they demonstrated that Membrane Attack Complex (MAC) triggered non-canonical NF-Kappa signaling through internalization into a Rab5 endosome. Here they further detail that this internalization involves ZFYVE21 that acts as a complement-induced Rab5 effector to regulate NF-Kappa signaling in endothelial cells. They further show that pharmacological inhibition of this pathway by administration of miltefosine blocks MAC-induced allograft vasculopathy in a humanized mouse model.

The authors are to be congratulated in presenting what is undoubtedly a tremendously elegant and comprehensive set of experiments. The experimental data are thorough and support their conclusions. I am, however, concerned about the relevance of their findings to clinical transplantation:

1. In both this paper and the previous *Jane-Wit Circulation*, the authors have developed a humanized model of allograft vasculopathy, which requires reconstitution of the immunodeficient murine host with human PBMCs. The vasculopathic process is thus T cell dependent. Presumably, in their model, the T cells that infiltrate the allograft recognise alloantigen via the direct pathway and bind to class II MHC xenoantigen on the human aortic graft? I think this raises an important question as to how relevant this process is for clinical transplantation. Although human endothelium can express MHC class II antigen (and could therefore act as a focus for direct-pathway CD4 T cell recognition), evidence supporting late direct-pathway CD4 T cell responses in clinical transplantation is remarkably sparse. Instead the work of Lechler and Suci-Foca instead suggests that the indirect-pathway CD4 T cell responses are likely to be more relevant. On that note, T cell infiltration of allograft neointima, as demonstrated in figure 5, is not a conspicuous feature of human allografts undergoing chronic rejection. Similarly, the work of Hancock and Turka has shown that alloantibody can mediate allograft vasculopathy in the absence of a recipient T cell population.

The clinical data included in their manuscript details synovium from patients with rheumatoid arthritis, which is thought to be a T cell dependent arthropathy. Although this potentially widens the scope of their findings, I would have preferred the inclusion of a similar analysis, but on either biopsies or explants from human allografts undergoing chronic rejection.

Reviewer #2 - expert in MAC (Remarks to the Author):

This manuscript describes an exhaustive study of the impact of exposure of endothelial cells *ex vivo* and *in vivo* to complement attack. The work includes an extremely detailed characterisation of the endosomal compartment generated during complement attack and the signalling complex leading to non-canonical NfKb activation that is formed. The identification of a key component of this complex enables the testing of a potential therapeutic approach to disrupt the signalling complex and reduce the inflammatory impact of complement attack. The work builds on published work from this group, particularly the 2015 paper (ref 1 here).

In all the studies the effects observed are ascribed to formation of the MAC; however, the evidence to support this assertion is missing. In both *in vitro* (EC in culture) and *in vivo* (human microvessels and coronary artery xenografts and SCID mice) the trigger is immune serum from highly sensitised transplant candidates supplemented with an unstated amount of fluor-labelled C9 added in to track MAC formation. As an aside, the source of C9 is described as "recombinant from Sigma"; this is an error as Sigma only supply plasma-derived human C9. Cells (or animals) are exposed to this immune serum (plus C9) to induce complement activation; endosomes are then isolated and characterised.

Proteomic differences between endosomes from control and complement-exposed cells are compared. The assumption that observed effects are MAC-mediated is based upon co-localisation of fluorescent C9 with Rab5 and other elements of the proposed signalling complex; other complement proteins (or indeed, other serum-derived proteomic hits) are not tested in this manner. In order to prove that the effects are indeed MAC-mediated it is essential to specifically prevent MAC formation! At its simplest this could be done by using sera depleted in individual terminal pathway components - easy to do in the *in vitro* system but complicated *in vivo* by the endogenous mouse complement system.

As it stands, the work convincingly shows an impact of exposing endothelial cells to serum (proof of dependence on the presence of antibody *in vitro* needs further work). It is likely that complement is an important effector but key evidence is lacking and there is no certainty that MAC is either necessary or sufficient for the observed effects.

Assuming that MAC can be firmly implicated with additional experimentation, then also missing is any evidence (or even hypothesis) of how MAC in this endosomal compartment acts as a nidus for a signalling complex. Does this require a MAC channel? Are there roles for the MAC inhibitor CD59?

Reviewer #3 - expert in phosphoinositides and endosomal trafficking (Remarks to the Author):

In this manuscript, Fang et al explore the molecular mechanisms underlying the proinflammatory actions of the classical complement during transplant organ rejection and connective tissue diseases in endothelial cells (ECs). This is a follow-up on previous work showing that Rab5 is required for membrane attack complex (MAC) signaling in targeted ECs after the internalization of complement. Here the authors identify the phosphatidylinositol-3-phosphate [PI(3)P]-binding protein ZFYVE21 as a complement-induced Rab5 effector that regulates non-canonical NF-kappaB signaling in ECs. They found that MAC induces an upregulation of ZFYVE21 via a post-translational stabilization that prevents its degradation by the proteasome. Remarkably, ZFYVE21 is required for the polyubiquitination of the PI(3,4,5)P3 3-phosphatase PTEN by the ubiquitin ligase SMURF2 and the proteasome-dependent degradation of the lipid phosphatase, resulting in increased PI(3,4,5)P3 levels on endosomes and recruitment of Akt and NF-kappaB-inducing kinase (NIK). The authors also claim that they deplete PI(3)P in vivo with miltefosine, a drug of dubious specificity, which reduces MAC-induced allograft vasculopathy. Finally, the authors find that ZFYVE21 induction occurs specifically in response to MAC and appears to be a specific biomarker for complement-mediated endothelial signaling in human synovial tissues.

Overall, this is an interesting study that clarifies the pro-inflammatory effects of complement in a non-canonical pathway leading to NF-kappaB activation, through the identification of a novel rab5 effector, ZFYVE21. While this study presents several novel and compelling findings, there are also a number of gaps that need to be filled in order for the data to be more definitive. These are as follows:

1. Generally, the number of technical and biological replicates should be consistently indicated in the legend to figures. The information provided is not always sufficient to evaluate the scientific rigor of this manuscript. A lot of key data is presented in the form of western blots, without any quantifications, which precludes the readers from evaluating the variability and strength of the data. This is not acceptable.
2. It may well be that the sera from allo-sensitized transplant candidates containing high titer panel reactive antibody (PRA) are known to induce antibody-mediated MAC assembly on human ECs. However, the study does not clearly show that ZFYVE21 is upregulated because of complement signaling. Can the authors use some inhibitors showing that MAC binding to EC membrane is required to achieve this effect?

3. It is unclear whether MAC induces Rab5 signaling through pore formation in EC membranes or whether it is simply taken up and internalized, mediating signals independently of membrane damage. If the latter applies, what is the cell surface receptor?
4. The isolation of Rab5/Mac+ vesicles by FACS is interesting, but the vesicles that are purified are very small, more likely to be endocytic vesicles than actual endosomes. This is surprising because a 45min pulse was applied and thus one would expect MAC to be delivered to rab5+ endosomes during this time. Why aren't larger (> 200 nm) Rab5+ endosomes isolated? The authors should use more standard endosome purification approaches to rule out technical artifacts related to FACS sorting that may prevent the isolation of large endosomes.
5. What is the fate of complement beyond its signaling effect. Is it ultimately targeted to lysosomes for degradation?
6. In Figure 1b, the experiment design does not separate the effect of the proteasome inhibitor (MG132) on ZFYVE21/NIK protein levels from that of PRA.
7. In Figure 1c, there is no information on how the experiment was conducted. Is this an anti-Rab5 immunoprecipitation from detergent extracts or an immunoisolation from detergent-free cell extracts? What was the starting material? FACS-purified endosomes? In general, the legends are a bit too brief and should contain more experimental detail.
8. In Figures 1e-f, the inhibitor which the authors use to block class 2 and 3 PI3Ks (KU-55993) is not very specific. There are now highly selective class 3 PI3K/Vps34 inhibitors, which dramatically deplete PI3P levels. Blocking Vps34 may be enough to reduce the effect of PRA, without contributions from class 2 PI3Ks. This should be tested.
9. In Figures 1l-m, it is unclear what PH domain is used: is it PI(3,4,5P)3-specific? What protein is it derived from? Some PH domains selectively recognize PI(4,5)P2, PI(3,4,5)P3, PI(3,4)P2 or even PI(3,4)P2/PI(3,4,5)P3, hence the need to be more specific. Importantly, the data does not look very convincing and the images are poor quality. Additional work must be conducted to validate these probes, which can be prone to a lot of artifacts. The FYVE domain probe should be largely soluble upon treatment with the Vps34 inhibitor.
10. Does ZFYVE21 physically interact with SMURF2?
11. The paper focuses on NF-kappaB signaling but there is no data on NF-kappaB activation/nuclear translocation. This should be investigated, at least in the ZFYVE21 knockdowns.
12. Miltefosine is not a very specific reagent, yet, it is used in two central figures of the paper. This is worrisome. If the purpose is to reduce PI(3)P levels, why not use a specific class 3 PI3K/Vps34 inhibitor both in vitro and in vivo? If the authors confirm the miltefosine results, it would give this referee a lot more confidence that the effects are real.

Reviewer #4 - expert in NF- κ B signalling (Remarks to the Author):

Activation of complement forming membrane attack complexes (MACs) on the surface of endothelial cells (ECs) plays an important role in pathology of transplant rejection. Classically MAC activation has been thought to cause endothelial cell death, however this group has been advancing the concept that MAC induces non-canonical NF κ B inflammatory signaling mediated by Rab5+ associated endosomes stabilizing the NF κ B inducing kinase (NIK).

In this paper, Caodi et al. use proteomics studies to identify the composition of activated complexes (MAC+/Rab5+) in FACS-sorted vesicles from antibody-activated ECs. From this analysis of 37 nm vesicles, the authors identify enrichment of 440 proteins. Reasoning that Rab5 activation involved proteins containing a PI3P lipid binding motif, and from the data indicating that ZFYVE21 is enriched in active endosomes and validate this finding by Western blot. ZFYVE21 is post-translationally stabilized from proteasomal degradation by Rab5 mediated recruitment, and binds PI3P. far western and co-IP data indicate that ZFYVE21 binds the GTP-bound, activated Rab5. ZFYVE21 displaces PTEN stabilizing PI(3,4,5)P₃, resulting in Akt recruitment. Investigation of mechanism of PTEN degradation, the authors demonstrate that SMURF2 ubiquitin ligase mediates PTEN degradation. The authors demonstrate the importance of the signaling pathway in inflammation in vivo.

Overall, this is a comprehensive, convincing study that extends these authors previous work. The integration of the proteomics, PI3 binding probes with protein interaction studies makes a compelling dissection of the Rab5 ZFYVE21pAKTNIK pathway. The HUVEC gel implant studies establish the pathway's relevance in vivo, although I am not sure whether HUVECs mimic primary ECs.

These are mostly minor issues in data availability and presentation.

The proteomics data need to be deposited in a public database, including the spectra and confidence of identification. This data should be included in the final manuscript. Also, the variance of the spectral counts of ZFYVE21 for each replicate should be shown in Supplemental Fig. S1.

The molecular weights of ZFYVE21, NIK, Akt, etc. should be displayed in the Western blots in Fig. 1. The page numbering overlies Fig. 1k, I cant evaluate the data demonstrating loss of NIK stabilization after ZFYVE21 knockdown.

I cant tell if Fig. 4a are ECs isolated from the subcutaneous implants, or whether these are HUVECs in vitro. I am suspecting this study is in vitro but this should be clarified. The Western blots in Fig. 4e are very faint and hard to discern.

The effect of Miltefosine on expression of NIK-dependent genes, such as SELE or VCAM1, should be confirmed in the images in vivo (Fig. 5).

Minor comments

Line 121, the abbreviation kDal is not standard. Line 270, the CFSE should be defined.

The "Ulex" staining shown in Fig. 5 should be described and interpreted.

Reviewers' comments:

Reviewer #1 - expert in allograft vasculopathy (Remarks to the Author):

This manuscript extends the authors' previous publications that have explored the contribution of complement and alloantibody to allograft vasculopathy.

In their previous work they demonstrated that Membrane Attack Complex (MAC) triggered non-canonical NF-Kappa signaling through internalization into a Rab5 endosome. Here they further detail that this internalization involves ZFYVE21 that acts as a complement-induced Rab5 effector to regulate NF-Kappa signaling in endothelial cells. They further show that pharmacological inhibition of this pathway by administration of miltefosine blocks MAC-induced allograft vasculopathy in a humanized mouse model.

The authors are to be congratulated in presenting what is undoubtedly a tremendously elegant and comprehensive set of experiments. The experimental data are thorough and support their conclusions. I am, however, concerned about the relevance of their findings to clinical transplantation:

1. In both this paper and the previous *Jane-Wit Circulation*, the authors have developed a humanized model of allograft vasculopathy, which requires reconstitution of the immunodeficient murine host with human PBMCs. The vasculopathic process is thus T cell dependent. Presumably, in their model, the T cells that infiltrate the allograft recognise alloantigen via the direct pathway and bind to class II MHC xenoantigen on the human aortic graft? I think this raises an important question as to how relevant this process is for clinical transplantation. Although human endothelium can express MHC class II antigen (and could therefore act as a focus for direct-pathway CD4 T cell recognition), evidence supporting late direct-pathway CD4 T cell responses in clinical transplantation is remarkably sparse. Instead the work of Lechler and Suci-Foca instead suggests that the indirect-pathway CD4 T cell responses are likely to be more relevant. On that note, T cell infiltration of allograft neointima, as demonstrated in figure 5, is not a conspicuous feature of human allografts undergoing chronic rejection. Similarly, the work of Hancock and Turka has shown that alloantibody can mediate allograft vasculopathy in the absence of a recipient T cell population. The clinical data included in their manuscript details synovium from patients with rheumatoid arthritis, which is thought to be a T cell dependent arthropathy. Although this potentially widens the scope of their findings, I would have preferred the inclusion of a similar analysis, but on either biopsies or explants from human allografts undergoing chronic rejection.

We thank the referee for his/her overall favorable opinion of our experimental contribution. One can debate the roles of direct or indirect pathway responses in late rejection. Most of the evidence in favor of the exclusive role of the indirect pathway, in which CD4 T cells are activated by host APC presentation of graft derived peptides, comes from mouse models in which graft APC, i.e. passenger leukocytes, are killed off shortly after transplantation, and the ability to sustain a direct response is lacking in these systems. Human grafts differ from rodent grafts in that they permanently retain MHC class II-expressing endothelium even after passenger leukocytes are killed and can continue to stimulate host CD4 T cells, either by direct or semi-direct pathways.

In support of the above, we previously found that 1) nanoparticle-mediated inhibition of MHC class II on donor endothelium results in immunologic ignorance, ablated T cell infiltration, and attenuated allograft vasculopathy,¹ and 2) PRA sera alone did not result in allograft vasculopathy in the absence of T cells (ref). Help to host B cells likely does involve indirectly activated CD4 T cells and our current hypothesis is that chronic rejection is a mixed response, involving antibody (plus complement) alteration of graft cells (notably graft endothelial cells) in a manner that continues to activate T cells by direct recognition. Indeed, human coronary vessels affected by vasculopathy show antibody and complement deposition on the endothelium² AND contain T cells (but not B cells) in a subendothelial position.^{3,4} Our humanized mouse model is admittedly biased towards direct allorecognition because "host" human myeloid cells are not effectively engrafted by adoptive transfer of PBMC. We believe these issues, including the limitations of our model, are best addressed in the Discussion and have added a paragraph to do so. *However, the current paper is not about vasculopathy per se, and this discussion does not directly bear upon our demonstration of the existence of a novel role for ZFYVE21 in the endosome-based signaling in the pathway by which MAC leads to endothelial activation.*

A second issue raised by the referee regards the use of RA and OA synovium to assess the relevance of this newly described mechanism. We differ in our opinion that RA is a pure T cell arthropathy. Many rheumatologists would agree that it is usually a mixed disorder, with anti-citrullinated peptide antibodies playing a significant role in most patients. However, our data show that even patients with RA lacking antibody and in OA patients, also lacking antibody, complement is still activated, MAC is deposited on endothelium, and

ZFYVE21 is stabilized. *We show these tissues to demonstrate that complement activation is associated with ZFYVE21 stabilization in a clinically relevant setting, and we believe the data support our point.* Finally, while an extensive survey of human samples displaying antibody-mediated allograft rejection is beyond the scope of the current study, we have added a limited number of samples to address the referee's request to document the activation of the MAC/ZFYVE21 pathway in this setting.

1. Cui, J., Qin, L., Zhang, J., Abrahimi, P., Li, H., Li, G., Tietjen, G.T., Tellides, G., Pober, J.S., Saltzman, M. Ex vivo pretreatment of human vessels with siRNA nanoparticles provides protein silencing in endothelial cells. *Nat Commun.* **8**, (2017). doi: 10.1038/s41467-017-00297-x.
2. Salomon RN, Hughes CC, Schoen FJ, Payne DD, Pober JS, Libby P. Human coronary transplantation-associated arteriosclerosis. Evidence for a chronic immune reaction to activated graft endothelial cells. *Am J Pathol.* **138**, 791-798 (1991).
3. Normann, S.J., Khan, S.R., Leelachaikul, P., Salomon, D.R. Origin of cells in the coronary intima during acute vascular rejection of the transplanted human heart. *J Heart Lung Transplant.* **11**, 492-499 (1992).
4. Rahmani, M., Cruz, R.P., Granville, D.J., McManus, B.M. Allograft vasculopathy versus atherosclerosis. *Circ Res.***99**, 801-815 (2006).

Reviewer #2 - expert in MAC (Remarks to the Author):

This manuscript describes an exhaustive study of the impact of exposure of endothelial cells ex vivo and in vivo to complement attack. The work includes an extremely detailed characterisation of the endosomal compartment generated during complement attack and the signalling complex leading to non-canonical NfKb activation that is formed. The identification of a key component of this complex enables the testing of a potential therapeutic approach to disrupt the signalling complex and reduce the inflammatory impact of complement attack. The work builds on published work from this group, particularly the 2015 paper (ref 1 here).

In all the studies the effects observed are ascribed to formation of the MAC; however, the evidence to support this assertion is missing. In both in vitro (EC in culture) and in vivo (human microvessels and coronary artery xenografts and SCID mice) the trigger is immune serum from highly sensitised transplant candidates supplemented with an unstated amount of fluor-labelled C9 added in to track MAC formation. As an aside, the source of C9 is described as "recombinant from Sigma"; this is an error as Sigma only supply plasma-derived human C9. Cells (or animals) are exposed to this immune serum (plus C9) to induce complement activation; endosomes are then isolated and characterised.

Proteomic differences between endosomes from control and complement-exposed cells are compared. The assumption that observed effects are MAC-mediated is based upon co-localisation of fluorescent C9 with Rab5 and other elements of the proposed signalling complex; other complement proteins (or indeed, other serum-derived proteomic hits) are not tested in this manner. In order to prove that the effects are indeed MAC-mediated it is essential to specifically prevent MAC formation! At its simplest this could be done by using sera depleted in individual terminal pathway components - easy to do in the in vitro system but complicated in vivo by the endogenous mouse complement system.

As it stands, the work convincingly shows an impact of exposing endothelial cells to serum (proof of dependence on the presence of antibody in vitro needs further work). It is likely that complement is an important effector but key evidence is lacking and there is no certainty that MAC is either necessary or sufficient for the observed effects.

Assuming that MAC can be firmly implicated with additional experimentation, then also missing is any evidence (or even hypothesis) of how MAC in this endosomal compartment acts as a nidus for a signalling complex. Does this require a MAC channel? Are there roles for the MAC inhibitor CD59?

We thank referee 2 for his/her appreciation of the extensive characterization of the work presented. In the Materials & Methods section, 'recombinant C9' has been changed to 'C9 protein purified from human serum (Sigma).' As stated in the Materials & Methods, 75µL of AlexaFluor 647-labeled C9 was added to 500µL PRA

and 1925 μ L gelatin veronal buffer and then added to IFN- γ pre-treated HUVEC carrying a Rab5- GFP reporter in a T175 flask at 37°C for 30 minutes.'

Our prior study showed that at the 30min timepoint when NIK was stabilized, MAC colocalized most frequently with Rab5+ vesicles containing EEA1 and APPL1, and less frequently with late Rab7+ endosomes or LAMP1+ phagosomes/phagolysosomes (ref 2 in the manuscript). Upon further IF analysis, we found that signaling components required for non-canonical NF- κ B including Akt, NIK, and IKK- α all colocalized with MAC+Rab5+ endosomes (ref 2 in the manuscript). Based on these data, we found it reasonable to presuppose that the MAC+Rab5+ compartment was a source enriched in undefined and relevant signaling components for proteomic analyses. To show this, we previously used sera fractionation of PRA sera to demonstrate that IgG alone, IgG plus C6-deficient sera, and IgG plus C9 deficient sera did not induce NIK-dependent inflammatory genes in human endothelial cells *in vitro* while IgG plus normal human sera did (ref 2 in the manuscript). Because C5b and C6 compose the backbone of the MAC pore (whereas C9 is required to expand its diameter), the inability of IgG+C6-deficient sera to elicit NIK-dependent gene formation would suggest that a functional MAC pore is required for pathway activation.

We have previously shown that PRA sera does not alter surface expression of complement inhibitors including CD46, CD55, and CD59 (ref 2 in the manuscript). A role for CD59 is possible in our studies and is an interesting direction for future work but is beyond the scope of the current studies which aims to define mechanisms of non-canonical NF- κ B activation by an endosome-associated mechanism.

As the referee notes, this approach cannot be simply tested in our mouse model as immunodeficient mice lacking C6 or C9 are not available. However, we had previously shown that neutralizing mouse C5 but not C5a did prevent Ig/MAC signaling in human endothelial cells *in vivo*. This work involved a collaboration with Alexion that had provided the antibody and we are unable to obtain additional sera to extend these findings to ZFYVE 21 stabilization. We hope the *in vitro* demonstration is sufficient to address the referee's concern.

Reviewer #3 - expert in phosphoinositides and endosomal trafficking (Remarks to the Author):

In this manuscript, Fang et al explore the molecular mechanisms underlying the proinflammatory actions of the classical complement during transplant organ rejection and connective tissue diseases in endothelial cells (ECs). This is a follow-up on previous work showing that Rab5 is required for membrane attack complex (MAC) signaling in targeted ECs after the internalization of complement. Here the authors identify the phosphatidylinositol-3-phosphate [PI(3)P]-binding protein ZFYVE21 as a complement-induced Rab5 effector that regulates non-canonical NF- κ B signaling in ECs. They found that MAC induces an upregulation of ZFYVE21 via a post-translational stabilization that prevents its degradation by the proteasome. Remarkably, ZFYVE21 is required for the polyubiquitination of the PI(3,4,5)P3 3-phosphatase PTEN by the ubiquitin ligase SMURF2 and the proteasome-dependent degradation of the lipid phosphatase, resulting in increased PI(3,4,5)P3 levels on endosomes and recruitment of Akt and NF- κ B-inducing kinase (NIK). The authors also claim that they deplete PI(3)P *in vivo* with miltefosine, a drug of dubious specificity, which reduces MAC-induced allograft vasculopathy. Finally, the authors find that ZFYVE21 induction occurs specifically in response to MAC and appears to be a specific biomarker for complement-mediated endothelial signaling in human synovial tissues.

Overall, this is an interesting study that clarifies the pro-inflammatory effects of complement in a non-canonical pathway leading to NF- κ B activation, through the identification of a novel rab5 effector, ZFYVE21. While this study presents several novel and compelling findings, there are also a number of gaps that need to be filled in order for the data to be more definitive. These are as follows:

1. Generally, the number of technical and biological replicates should be consistently indicated in the legend to figures. The information provided is not always sufficient to evaluate the scientific rigor of this manuscript. A lot of key data is presented in the form of western blots, without any quantifications, which precludes the readers from evaluating the variability and strength of the data. This is not acceptable.

As requested we have significantly revised and expanded all of the figure legends to provide more information regarding the replications of each experiment, and we have performed densitometry quantifications on all lanes

of virtually all Western blots in the study as shown below:

2. It may well be that the sera from allo-sensitized transplant candidates containing high titer panel reactive antibody (PRA) are known to induce antibody-mediated MAC assembly on human ECs. However, the study does not clearly show that ZFYVE21 is upregulated because of complement signaling. Can the authors use some inhibitors showing that MAC binding to EC membrane is required to achieve this effect?

We previously performed sera fractionation experiments using PRA sera to demonstrate that MAC but not anaphylatoxins or IgG binding were responsible for NIK-dependent genes. We extended these studies to include biochemical assessments of ZFYVE21, pAkt, and NIK as shown in Fig 1c. Additional sentences were added to the Introduction, and The Results and Figure Legends have also been appropriately revised to reflect these findings.

3. It is unclear whether MAC induces Rab5 signaling through pore formation in EC membranes or whether it is simply taken up and internalized, mediating signals independently of membrane damage. If the latter applies, what is the cell surface receptor?

We previously used sera fractionation of PRA sera to demonstrate that IgG alone, IgG plus C6-deficient sera, and IgG plus C9 deficient sera did not induce NIK-dependent inflammatory genes in human endothelial cells *in vitro* while IgG plus normal human sera did (ref 2 in the manuscript). Because C5b-C6-C7 compose the backbone of the MAC pore (whereas C8-C9 is required to expand its diameter), the inability of IgG plus C6-deficient sera to elicit NIK-dependent genes would suggest that a functional MAC pore is required for pathway activation. This notion is supported by the observation that pore-forming molecules like melittin and ionomycin similarly upregulated NIK (ref 2 in the manuscript).

We previously reported that we do not see evidence of irreversible membrane damage in cells treated with PRA as assessed by exclusion of propidium iodide (ref 2 in the manuscript). However, we have not initiated studies to identify markers of membrane damage which may serve as a trigger for MAC internalization which is outside the scope of the current manuscript which seeks to define mechanism(s) underlying how Rab5 activity could mediate pAkt recruitment to MAC+Rab5+ endosomes.

We also have shown that NIK stabilization is prevented by blocking clathrin-mediated endocytosis (ref 1 in the manuscript). As requested, we have repeated these experiments to show that the same applies to ZFYVE21 stabilization in Fig 1 e. The identity of a cell surface receptor or cofactor mediating MAC internalization is an interesting question worthy of examination in future studies, but again is outside the scope of the current manuscript. Identifying molecule(s) related to MAC internalization which may include oxidized/damaged phospholipids or perhaps flippase or scramblase and/or an unidentified receptor/cofactor(s) are all interesting questions but again are outside the scope of the current study.

4. The isolation of Rab5/Mac+ vesicles by FACS is interesting, but the vesicles that are purified are very small, more likely to be endocytic vesicles than actual endosomes. This is surprising because a 45min pulse was applied and thus one would expect MAC to be delivered to rab5+ endosomes during this time. Why aren't larger (> 200 nm) Rab5+ endosomes isolated? The authors should use more standard endosome purification approaches to rule out technical artifacts related to FACS sorting that may prevent the isolation of large endosomes.

We chose a FACS gating strategy in an attempt to purify Rab5+ early endosomes. Our prior study showed that at the 30min timepoint when NIK was stabilized, MAC colocalized most frequently with this Rab5+ vesicles containing EEA1 and APPL1, and less frequently with late Rab7+ endosomes or LAMP1+ phagosomes/phagolysosomes (ref 1 in the manuscript). These data indicated MAC colocalization with Rab5+ early endosomes. As early endosomes are smaller in size than other compartments containing docked or residual Rab5 proteins including late endosomes, phagolysosomes, ER, Golgi, and mitochondria, we intentionally limited the size of FACS-sorted endosomes for proteomic analyses to smaller vesicular sizes in an effort to capture early Rab5+ endosomes and exclude these larger membrane-bound subcellular structures. Contaminating proteins from these structures are typically found using conventional endosome isolation methods such as sucrose gradients which is why these approaches were not used, though reviewer #3 is correct in pointing out that intentional exclusion of large endosomes may have occurred using our current approach. In Fig S1b, there were in fact sorted events >200nm, but the overall diameter of MAC+Rab5+ endosomes was $112\pm 20\text{nm}$.

We admit that some component “scrambling” could occur during sonication and we only draw conclusions about specific proteins that can be validated by morphological assessment and co-immunoprecipitation from intact cells. It is also possible that the smaller size distribution of our endosome preparations was influenced by the use of sonication which is a second limitation of our strategy. We have added the Discussion:

5. What is the fate of complement beyond its signaling effect. Is it ultimately targeted to lysosomes for degradation?

This is an interesting question which may be relevant to termination of the endosome signaling pathway. We have shown colocalization of C9 using immune electron microscopy within double-membraned structures morphologically resembling autophagosomes (unpublished data, see image), and confocal images showed colocalization of MAC staining with LAMP1+ structures, which we published previously (ref 2 in the manuscript).

Such images can be added to the supplementary data, but further analyses of this process go well beyond the scope of the present study which was not intended to identify degradative route(s) of MAC.

6. In Figure 1b, the experiment design does not separate the effect of the proteasome inhibitor (MG132) on ZFYVE21/NIK protein levels from that of PRA.

Figure 1b was mislabeled as including 'PRA' and in fact only included MG132. We are sincerely appreciative that the reviewer has noticed this error. The figure has been revised to reflect the fact that only MG132 was used in the study.

7. In Figure 1c, there is no information on how the experiment was conducted. Is this an anti-Rab5 immunoprecipitation from detergent extracts or an immunoisolation from detergent-free cell extracts? What was the starting material? FACS-purified endosomes? In general, the legends are a bit too brief and should contain more experimental detail.

In Fig 1c, HUVEC were lysed using SDS-free RIPA buffer prior to antibody co-immunoprecipitation with protein A/G agarose beads. This protocol was applicable to all co-immunoprecipitation studies performed in the current manuscript and was described in detail in the Materials & Methods in reference 2, which was cited in brief in the Materials & Methods of the current manuscript. We apologize for the lack of clarity which could have been confusing to the reviewer. We have significantly expanded all of the figure legends and included additional material in the Materials & Methods section to address the referee's request.

8. In Figures 1e-f, the inhibitor which the authors use to block class 2 and 3 PI3Ks (KU-55993) is not very specific. There are now highly selective class 3 PI3K/Vps34 inhibitors, which dramatically deplete PI3P levels. Blocking Vps34 may be enough to reduce the effect of PRA, without contributions from class 2 PI3Ks. This should be tested.

Complying with the reviewer's request, we assessed two selective class 3 PI3K inhibitors, SAR405 and PIKIII as well as KU-55933, and confirmed that, to varying degrees, that these drugs blocked C9+Rab5+FYVE+ MFI, induction of ZFYVE21, NIK-dependent genes, and EC-mediated T cell activation in Fig 4 at doses which were previously reported to be selective for PI(3)P lipid depletion. As mentioned in the text, at doses reported in the literature conferring class 3 PI3K selectivity (~5nM), SAR405 did not show PI(3)P depletion in HUVEC unlike PIKIII. However, at higher doses, SAR405 (2.5uM) significantly attenuated a panel of 5 NIK-dependent inflammatory genes as did PIKIII (2.5µM) and KU55933 (20µM, data not shown).

9. In Figures 1l-m, it is unclear what PH domain is used: is it PI(3,4,5P)3-specific? What protein is it derived from? Some PH domains selectively recognize PI(4,5)P2, PI(3,4,5)P3, PI(3,4)P2 or even PI(3,4)P2/PI(3,4,5)P3, hence the need to be more specific. Importantly, the data does not look very convincing and the images are poor quality. Additional work must be conducted to validate these probes, which can be prone to a lot of artifacts. The FYVE domain probe should be largely soluble upon treatment with the Vps34 inhibitor.

The PH-RFP construct as cited in the Materials & Methods was obtained as a gift from Dr. Pietro de Camilli at Yale. The construct contains the PH domain from Akt1, whose selective binding to PI(3,4,5)P3 has been heavily described in the literature. The description of the probe was inserted into the Materials & Methods section and the relevant references have been inserted in the text. The PH-RFP probe was validated by showing a dose-dependent decrease of PH MFI following pan-PI3K inhibition with LY294002 (Fig S2c) and correlated with confocal analyses (Fig S2d). The PH reporter was further internally validated in our study by showing that PH-RFP probe MFI correlated biochemically with increased levels of pAkt in Rab5 co-IPs with PRA and decreased levels of pAkt in Rab5 co-IPs with PRA treatment of ZFYVE21 siRNA EC (Fig 1l vs Fig 1n,o). As above, many research groups have utilized and validated the specificity of this probe in numerous assays, most notably, confocal microscopy, which we showed to have moderate correlations using the PH-RFP construct in FACS analyses.

The 2XFYVE-GFP probe as cited in the Materials & Methods section was also obtained as a gift from the de Camilli lab at Yale. This construct contains the FYVE domain from HRS in tandem, and has been extensively

characterized in the literature as binding specifically to PI(3)P. We have inserted the appropriate references, and we have internally validated this probe by showing that FYVE-GFP probe MFIs increases with biochemical colocalization of ZFYVE21 to Rab5+ endosomes (Fig 4e vs Fig S2b) and FYVVE-GFP probe MFI decreases following PI(3)P depletion with PI(3)K depletion with KU-55933, a treatment that blocked ZFYVE21 recruitment to Rab5+ endosomes (Fig S2c vs Fig 1g). As above, correlations between the FYVE-GFP probe and concurrent confocal analyses were performed (Fig S2d). We have strengthened FYVE-GFP probe correlations with selective PI(3)K inhibitors, SAR405 and PIKIII per the reviewer's request. We have included a description of the 2XFYVE-GFP probe above in the Materials & Methods.

10. Does ZFYVE21 physically interact with SMURF2?

In cell-free binding assays *in vitro*, we observed that ZFYVE21 does not physically interact with SMURF2 as shown in Fig 3e.

11. The paper focuses on NF-kappaB signaling but there is no data on NF-kappaB activation/nuclear translocation. This should be investigated, at least in the ZFYVE21 knockdowns.

As requested, we have added experiments showing that RelB nuclear translocation is inhibited following PRA treatment in ZFYVE21 knockdowns in Fig 1m.

12. Miltefosine is not a very specific reagent, yet, it is used in two central figures of the paper. This is worrisome. If the purpose is to reduce PI(3)P levels, why not use a specific class 3 PI3K/Vps34 inhibitor both *in vitro* and *in vivo*? If the authors confirm the miltefosine results, it would give this referee a lot more confidence that the effects are real.

Miltefosine was selected for testing *in vivo* because 1) it is an FDA-approved drug with known, annotated effects *in vivo*, and 2) miltefosine has a long half-life >30dy, such that repeat drug administration is not required over time, bypassing confounding effects of administering miltefosine in the presence of circulating lymphocytes, thus isolating its effects entirely on graft tissues. Unfortunately, due to the only recent discovery of many of these selective molecules, we are not aware of any specific class 3 PI3K/Vps34 inhibitors that meet these criteria. We have included additional data analyzing effects of 2 class 3 PI3K/Vps34 inhibitors, SAR450 and PIKIII, *in vitro* with regards to ZFYVE21 signaling, EC activation, and EC-mediated priming of alloimmune CD4+T cells but are unable to find *in vivo* studies validating PI(3)P depleting effects of these or any class 3 PIK3-specific drug inhibitors.

If such drugs existed, testing of such inhibitors would nevertheless require multiple drug administrations with potential and unknown confounding effects on one or more immune cell populations. For example, mice with dendritic cell-specific deletion of Vps34 show deficits in antigen cross-presentation (ref below). Comprehensive mass spectrometry assessments to determine half lives of PI3K/Vps34 inhibitors, testing *in vivo* routes of clearance, and testing its actions on multiple cell types including EC and circulating immune cells are important questions that may lead to novel therapies in patients, but is well beyond the scope of the current manuscript which aims to define novel biochemical mechanisms and molecules involved in endosome-associated MAC signaling and show that these molecules are involved in EC activation.

Vps34 deficiency is embryonically lethal and tissue-specific deletions of Vps34 are emerging. As such we do not wish to imply in our study that systemic PI3K/Vps34 inhibition is a singular and/or ideal method for inhibiting complement-mediated disease. Rather we wish to demonstrate as proof of principle that a novel Rab5 effector, ZFYVE21, can represent a potentially druggable target. Alternative and perhaps safer and more viable methods for inhibition of ZFYVE21 and/or other pathway-associated molecules may emerge from ongoing mechanistic studies surrounding this molecule. We have added additional sentences in the last paragraph of the Discussion to clarify any misunderstanding emerging from our mechanistic pathway studies which were again intended to define how Rab5 activity could regulate pAkt recruitment to MAC+Rab5+ endosomes to elicit EC activation.

In light of the above, we hope the reviewer will not demand we repeat the extremely costly and time consuming *in vivo* experiments. We do believe the reviewers' concerns are valid and important and should be tested in separate studies which go well beyond the scope of the current manuscript. As mentioned, we have analyzed 2 PI3K/Vps34 inhibitors *in vitro* which may form the basis for these future *in vivo* studies.

Parekh, V.V., Pabbisetty, S.K., Wu, L., Sebzda, E., Martinez, J., Zhang, J., Van Kaer, L. Autophagy-related protein Vps34 controls the homeostasis and function of antigen cross-presenting CD8 α + dendritic cells. *Proc Natl Acad Sci.* **114**, doi: 1073/pnas.1706504114.

Reviewer #4 - expert in NF-kB signaling (Remarks to the Author):

Activation of complement forming membrane attack complexes (MACs) on the surface of endothelial cells (ECs) plays an important role in pathology of transplant rejection. Classically MAC activation has been thought to cause endothelial cell death, however this group has been advancing the concept that MAC induces non-canonical NFkB inflammatory signaling mediated by Rab5+ associated endosomes stabilizing the NFkB inducing kinase (NIK).

In this paper, Caodi et al. use proteomics studies to identify the composition of activated complexes (MAC+/Rab5+) in FACS-sorted vesicles from antibody-activated ECs. From this analysis of 37 nm vesicles, the authors identify enrichment of 440 proteins. Reasoning that Rab5 activation involved proteins containing a PI3P lipid binding motif, and from the data indicating that ZFYVE21 is enriched in active endosomes and validate this finding by Western blot. ZFYVE21 is post-translationally stabilized from proteasomal degradation by Rab5 mediated recruitment, and binds PI3P. far western and co-IP data indicate that ZFYVE21 binds the GTP-bound, activated Rab5. ZFYVE21 displaces PTEN stabilizing PI(345)P, resulting in Akt recruitment. Investigation of mechanism of PTEN degradation, the authors demonstrate that SMURF2 ubiquitin ligase mediates PTEN degradation. The authors demonstrate the importance of the signaling pathway in inflammation *in vivo*. Overall, this is a comprehensive, convincing study that extends these authors previous work. The integration of the proteomics, PI3 binding probes with protein interaction studies makes a compelling dissection of the Rab5 ZFYVE21pAKTNIK pathway. The HUVEC gel implant studies establish the pathway's relevance *in vivo*, although I am not sure whether HUVECs mimic primary ECs. These are mostly minor issues in data availability and presentation.

The proteomics data need to be deposited in a public database, including the spectra and confidence of identification. This data should be included in the final manuscript. Also, the variance of the spectral counts of ZFYVE21 for each replicate should be shown in Supplemental Fig. S1.

We thank the reviewer for the encouraging comments. The proteomics datasets have been deposited in the Yale Protein Expression Database which can be publicly accessed using the following link: <https://medicine.yale.edu/keck/proteomics/yped/>. This has been added to the revised Materials & Methods section.

The molecular weights of ZFYVE21, NIK, Akt, etc. should be displayed in the Western blots in Fig. 1. The page numbering overlies Fig. 1k, I cant evaluate the data demonstrating loss of NIK stabilization after ZFYVE21 knockdown.

We have added the molecular weights for these molecules in Fig 1. The culprit figure is now 'Fig I' and the page numbering in the revised manuscript does not appear to overlie the Western blot. We apologize for this formatting error which did not appear on our submitted Word document containing the figure images.

I can't tell if Fig. 4a are ECs isolated from the subcutaneous implants, or whether these are HUVECs *in vitro*. I am suspecting this study is *in vitro* but this should be clarified. The Western blots in Fig. 4e are very faint and hard to discern.

The ECs used in Fig 4a were HUVEC cultured *in vitro*, and the Figure Legend has been updated for clarification. The Western blot in Fig 4e is now Fig 4h and the contrast on the bands have been enhanced using a higher resolution.

The effect of Miltefosine on expression of NIK-dependent genes, such as SELE or VCAM1, should be confirmed in the images *in vivo* (Fig. 5).

We have added this data in Fig 5b to show E-selectin inhibition in Ulex+ intimal regions following miltefosine treatment.

Minor comments

Line 121, the abbreviation kDal is not standard. Line 270, the CFSE should be defined. The “Ulex” staining shown in Fig. 5 should be described and interpreted.

‘kDal’ has been changed to ‘kD’ throughout the manuscript. The following text defining ‘CFSE’ and describing ‘Ulex’ have been added to the Results section of the text:

“To test the functional effects of inflammatory gene induction, we co-cultured PRA-treated EC containing control siRNA or ZFYVE21 siRNA with CD4+ T cells labeled with CFSE (carboxyfluorescein succinimidyl ester) to monitor cell proliferation. In this system, human EC function as antigen presenting cells to directly induce proliferation of allogeneic CD4+ T cells assayed by dilution of CFSE fluorescence.”

“Compared to IgG-depleted PRA controls, IgG+ PRA-treated xenografts showed increased intimal staining for mouse C4d, a marker of complement activation, ZFYVE21, SMURF2, and NIK, markers which colocalized with Ulex europaeis (Ulex), a lectin binding human EC (Fig 5a).”

** See Nature Research's author and referees' website at www.nature.com/authors for information about policies, services and author benefits

This email has been sent through the Springer Nature Tracking System NY-610A-NPG&MTS

Confidentiality Statement:

This e-mail is confidential and subject to copyright. Any unauthorised use or disclosure of its contents is prohibited. If you have received this email in error please notify our Manuscript Tracking System Helpdesk team at <http://platformsupport.nature.com>.

*Details of the confidentiality and pre-publicity policy may be found here <http://www.nature.com/authors/policies/confidentiality.html>
Privacy Policy | Update Profile*

Reviewer #1 (Remarks to the Author):

No concerns. The authors have addressed my comments.

Reviewer #2 (Remarks to the Author):

The manuscript is improved in review and most of my concerns are addressed in part. I do however, have some important residual concerns that are not adequately addressed.

1. I queried the C9 dosage and am now told that 75ul of a stock that I calculate to be 4mg/ml (if no loss in labelling/ concentration - not tested) in PBS is added to 500ul PRA serum then diluted with 1925ul GVB. This equates to ~300ug C9 added to 500ul serum, approximately ten-fold physiological. Why is this enormous dose needed in the experiment? Were aggregates of fluorescent C9 (inevitable at these high concentrations) eliminated? If not, how can the authors be sure that fluorescence in endosomes is not just C9 aggregates?

2. As proof of dependence on MAC the authors refer to previous work where they have tested in vitro C6- or C9-depleted (state deficient?) sera do not induce inflammasome activation (as reported by others for MAC triggering of inflammasome); given the importance of the question, why are such controls (plus adding back the missing component) not included here? The suggestion that C5b and C6 are "the backbone" of the MAC pore" with C9 merely required "to expand its diameter" is simply wrong and demonstrates a lack of current understanding of MAC structure - there is no pore at the pre-C9 stages!!

3. The authors state that the dependence of observed effects on MAC cannot be tested in vivo because "immunodeficient mice lacking C6 or C9 are not available". C5 deficient mice are widely available, C6-deficient mice have been around for at least a decade and widely used, and C7-deficient mice are more recently available. There is really no excuse for not testing in one of these lines.

Reviewer #4 (Remarks to the Author):

This is a comprehensive, mechanistic study that advances the mechanisms of inflammatory endothelial signaling through the non-canonical NFkB pathway initiated by the MAC.

Reviewer #5 - in replacement of reviewer #3 (Remarks to the Author):

I have mainly considered the response of the authors to the points raised by reviewer 3. Overall I was not very convinced by the response to the points raised by this reviewer.

Comments to specific points:

1. The authors now present, as figures for reviewers, quantifications of the western blots shown in the manuscript. However, they do not present quantifications based on multiple experiments, with error bars.
2. The authors have not used inhibitors showing that MAC binding to EC membrane, as requested by the reviewer.
3. The author response to this point is satisfactory.
4. I agree with reviewer 3 that the sizes of the isolated Rab5+ and MAC+ vesicles are incompatible with endosomes and would be more likely to represent endocytic vesicles (which have a size of 100 nm) or fragmented endosomes. Endosomes typically have a size of 500-1000 nm, i.e., much larger than the sizes of the isolated vesicles. The author response on this point is not convincing, and this crucial point calls for high-resolution fluorescence microscopy (with appropriate quantifications) which should demonstrate clearly whether MAC and ZFYVE21 localize to Rab5 positive endosomes, and whether this depends on PI(3)P.
5. Even though the authors were unable to give a clear-cut answer to the reviewer question, I think their reply here is satisfactory given that this point is not central to the message of the manuscript.
6. This point has been addressed by the authors.
7. This point has been addressed by the authors.

8. It is a little weird that such high concentrations of SAR405 were required, but otherwise the authors have addressed this point.

9. It is not correct that the Akt PH domain is specific for PI(3,4,5)P3. It also binds to PI(3,4)P2.

10. The lack of interaction between ZFYVE21 and SMURF2 begs the question of how SMURF2 is activated by ZFYVE21.

11. This point has been addressed by the authors.

12. I completely agree with reviewer 3 in that miltefosine is not a specific inhibitor of class III PI3K. I sympathize with the authors that it would be a tall order to repeat all the in vivo experiments with a more specific inhibitor. However, the authors need to strongly tone down their conclusions based on this inhibitor. For instance, the current version of the abstract claims that PI(3)P depletion reduces ZFYVE21 induction, EC activation and MAC-induced allograft vasculopathy in a humanized mouse model. This is a strong overstatement since the results might well be explained by other effects of miltefosine.

Reviewer #1 (Remarks to the Author):

No concerns. The authors have addressed my comments.

Reviewer #2 (Remarks to the Author):

The manuscript is improved in review and most of my concerns are addressed in part. I do however, have some important residual concerns that are not adequately addressed.

1. I queried the C9 dosage and am now told that 75ul of a stock that I calculate to be 4mg/ml (if no loss in labeling/ concentration - not tested) in PBS is added to 500ul PRA serum then diluted with 1925ul GVB. This equates to ~300ug C9 added to 500ul serum, approximately ten-fold physiological. Why is this enormous dose needed in the experiment? Were aggregates of fluorescent C9 (inevitable at these high concentrations) eliminated? If not, how can the authors be sure that fluorescence in endosomes is not just C9 aggregates? **Large concentrations were required as the labeling process was inefficient, leading to loss. Prior to testing, we calculated the percent of labeled C9 and found using the provided equations in the manufacturer's instructions (Molecular Probes, cat #A20173) that the moles AF647 dye incorporated per mole C9 protein ranged between 0.52-1.21. The optimal dye labeling per the manufacturer's instructions was 3-7 moles of AF647 dye per mole of protein. We tried different concentrations of C9 ranging from 200ug-800ug and found that at concentrations between 600-800ug, in confocal microscopy studies the number of C9+Rab5+ endosomes in Rab5-GFP HUVEC treated with PRA supplemented with C9-AF647 approximated that of native C9+Rab5+ endosomes stained by Abs. We thus used higher concentrations of C9-AF647 for the proteomic studies.**

To address the reviewer's concerns regarding C9 aggregates causing ZFYVE21 stabilization, we treated HUVEC with C9-AF647 at the same concentration used for proteomic analyses (lane 4) and found that C9-AF647 alone did not induce ZFYVE21 (left). However C9-AF647, when combined with IgG+ plus C6- IgG-, was able to rescue ZFYVE21 induction, indicating that terminal complement activation incorporating C9-AF647 was required for ZFYVE21. These studies effectively ruled out the possibility that C9 aggregates alone could induce ZFYVE21. It should be noted that C9-AF647 was only used for the proteomic studies and NOT for the remainder of the paper where unsupplemented PRA sera was used to study ZFYVE21. As such, it is impossible that ZFYVE21 induction could have occurred secondary to exogenous C9 aggregates. Additionally, as shown on the right below treatment of HUVEC for 30min with C9-647 resulted in significantly lower numbers of C9+Rab5+ vesicles by I.F. analysis vs PRA-treated HUVEC stained for C9 (red) and Rab5 (green). Together, these data below show that C9-AF647 aggregates did not induce ZFYVE21 nor did it form substantial numbers of C9+Rab5+ endosomes.

2. As proof of dependence on MAC the authors refer to previous work where they have tested *in vitro* C6- or C9-depleted (state deficient?) sera do not induce inflammasome activation (as reported by others for MAC triggering of inflammasome); given the importance of the question, why are such controls (plus adding back the missing component) not included here? The suggestion that C5b and C6 are "the backbone" of the MAC pore" with C9 merely required "to expand its diameter" is simply wrong and demonstrates a lack of current understanding of MAC structure - there is no pore at the pre-C9 stages!! **There is evidence of both functional (Farkas I, *J Physiol*, 2002 Mar 1;539(Pt 2):537-45) and anatomical (Sharp TH, et al. *Cell Rep*. 2016 Apr 5;15(1);1-8, Aleshin AE, *J Biol Chem*. 2012. Mar 23;287(13):10210-22., Zalman LS, *Mol Immunol*. 1990 Jun;27(6):533-7) pore formation by C5b-8. However, we have made textual corrections to the paper to satisfy the reviewer's concerns and will take into consideration the above moving forward.**

3. The authors state that the dependence of observed effects on MAC cannot be tested *in vivo* because "immunodeficient mice lacking C6 or C9 are not available". C5 deficient mice are widely available, C6-deficient mice have been around for at least a decade and widely used, and C7-deficient mice are more recently available. There is really no excuse for not testing in one of these lines. **In the current manuscript we have performed *in vitro* studies using HUVEC where PRA sera were separated into IgG+ and IgG- components and used in recombination experiments in conjunction with C6-deficient sera (Fig 1c). This experiment was performed identically to that in a previously published paper (Jane-wit, et al. *Circulation* 2013 Dec 3;128(23):2504-16) and showed that MAC and not IgG+ (DSA) or anaphylatoxins were responsible for upregulating ZFYVE21 and NIK.**

We have previously published a manuscript demonstrating that an anti-C5 Ab binding to murine C5 can block MAC, NIK, EC activation, and allograft vasculopathy *in vivo* in response to complement activation secondary to alloantibody or ischemia reperfusion injury (Qin, et al. 2016. *Am J Transplant*). In the future, as the reviewer aptly states, similar studies using C6 or C9 immunodeficient mice will be required to definitively show that the presently studied pathway is MAC-dependent *in vivo*. These studies are outside the scope of the current manuscript which identifies a new Rab5 effector, ZFYVE21, defines an endosome-based signaling mechanism by which it operates, and demonstrates its relevance *in vivo* and in patient tissues.

Reviewer #4 (Remarks to the Author):

This is a comprehensive, mechanistic study that advances the mechanisms of inflammatory endothelial signaling through the non-canonical NFkB pathway initiated by the MAC.

Reviewer #5 - in replacement of reviewer #3 (Remarks to the Author):

I have mainly considered the response of the authors to the points raised by reviewer 3. Overall I was not very convinced by the response to the points raised by this reviewer.

Comments to specific points:

1. The authors now present, as figures for reviewers, quantifications of the western blots shown in the manuscript. However, they do not present quantifications based on multiple experiments, with error bars.

The Western blot pooled densitometry is shown below. N numbers indicating experimental replicates are listed below and are also stated in the Figure Legends in the text. As we typically use multiple donors to obtain HUVEC as well as multiple pooled batches of human PRA sera to reproduce our data, it is expected that there will be variability between experiments. The general trends as can be seen are reproducible but with wide variability in certain treatment groups.

2. The authors have not used inhibitors showing that MAC binding to EC membrane, as requested by the reviewer. As discussed in our correspondence with Dr. Gebala, we too are not aware of any pharmacologic MAC inhibitors. If such an inhibitor were available, we would be more than happy to incorporate this reagent into our future studies. Studies involving CD59 or other endogenous inhibitors of complement/MAC are an important area of investigation which are the focus of ongoing studies in the lab. However, we feel that these data are outside the scope of the current study which has identified a new Rab5 effector, demonstrated its mechanism of action, and showed its relevance *in vivo* and in patient samples.

3. The author response to this point is satisfactory.

4. I agree with reviewer 3 that the sizes of the isolated Rab5+ and MAC+ vesicles are incompatible with endosomes and would be more likely to represent endocytic vesicles (which have a size of 100 nm) or fragmented endosomes. Endosomes typically have a size of 500-1000 nm, i.e., much larger than the sizes of the isolated vesicles. The author response on this point is not convincing, and this crucial point calls for high-resolution fluorescence microscopy (with appropriate quantifications) which should demonstrate clearly whether MAC and ZFYVE21 localize to Rab5 positive endosomes, and whether this depends on PI(3)P.

We have performed dual immune-electron microscopy and have quantified the average diameters of intact C9+Rab5+ endosomes in cells. This analysis shown in Supplemental Fig 1b with quantifications revealed that the MAC+Rab5+ compartment following PRA treatment had an average size of 1062nm vs 670nm in Rab5+ vesicles. The increased size distribution of MAC+Rab5+ endosomes vs Rab5+ endosomes via immune-electron microscopy both mirrors our observation in sorted vesicles and is congruent in size with an endosome population. As mentioned in the Discussion section, after sonication, and intentional gating on smaller event sizes, the final C9+Rab5+ population showed a smaller average size than intact endosomes as quantified by immune-electron microscopy.

5. Even though the authors were unable to give a clear-cut answer to the reviewer question, I think their reply here is satisfactory given that this point is not central to the message of the manuscript.

6. This point has been addressed by the authors.

7. This point has been addressed by the authors.

8. It is a little weird that such high concentrations of SAR405 were required, but otherwise the authors have addressed this point.

9. It is not correct that the Akt PH domain is specific for PI(3,4,5)P3. It also binds to PI(3,4)P2. **This fact has been added to the Materials & Methods section of the manuscript.**

10. The lack of interaction between ZFYVE21 and SMURF2 begs the question of how SMURF2 is activated by ZFYVE21. **SMURF2 recruitment to the signaling endosome population is an ongoing point of investigation in the lab for which we do not have sufficient data to address at the current time. This may represent an important area of focus for a future manuscript.**

11. This point has been addressed by the authors.

12. I completely agree with reviewer 3 in that miltefosine is not a specific inhibitor of class III PI3K. I sympathize with the authors that it would be a tall order to repeat all the *in vivo* experiments with a more specific inhibitor. However, the authors need to strongly tone down their conclusions based on this inhibitor. For instance, the current version of the abstract claims that PI(3)P depletion reduces ZFYVE21 induction, EC activation and MAC-induced allograft vasculopathy in a humanized mouse model. This is a strong overstatement since the results might well be explained by other effects of miltefosine. **The vocabulary in the abstract and results sections have been changed as highlighted in red text in the revised manuscript. We chose to embark on a bioinformatics search rather than use known specific PI(3)P inhibitors *in vivo*. This is because all specific PI(3)P inhibitors to date have not been tested *in vivo* and thus have unknown toxic profiles and half-lives, thus necessitating repeated dosing to possibly ensure a phenotypic effect. Such repeated dosing would create nonspecific drug exposure to immune cells that would complicate interpretation of effects on allograft vasculopathy. Miltefosine, on the other hand, though possibly nonspecific has a well-documented half-life exceeding the total time of our *in vivo* studies, and thus allowed us to use a dosing regimen that only exposed the artery xenografts to the drug.**

Reviewers' comments:

Reviewer #2 (Remarks to the Author):

This is the third time I have reviewed this manuscript. I will restrict my comments to the responses (and changes) made in addressing my last review critique:

1. I asked for clarification regarding the dose of C9 used, the amount of label incorporated and whether there was significant aggregation of the C9 post-labelling given the propensity of this protein to aggregate. The majority of the response given does not address these questions. No analysis of labelling density or efficiency is made, instead the authors refer to "manufacturer's instructions" - this is not at all helpful. No attempt is made to test whether the C9 is aggregated which would not only reduce its function but also increase the likelihood of "non-specific" uptake into the cells. I do not understand why the authors cannot address the points made.
2. I asked why the authors did not here use C6- and/or C9-depleted/deficient sera (and addbacks) as controls, and ask for clarification that the MAC pore includes multiple copies of C9. The first question is ignored, the second addressed by a list of papers purported to support the contention that C5b-8 is a pore. The structural composition of the MAC pore is made very clear from recent structural reports. C5b-8 may disrupt lipid in some context but it CANNOT form a pore. This needs to be clear in the text to avoid propagating misunderstandings.
3. I asked why the authors do not test in vivo in deficient mice (that they wrongly stated were not available). Most of the response is irrelevant to the question asked. Only in the last two sentences is the question addressed with the statement that "In the future studies using C6 or C9 immunodeficient mice will be required to definitively show that the presently studied pathway is MACdependent these studies are outside the scope of the current manuscript". As I stated before, there is really no excuse for not testing in one of these lines. This is particularly the case for publication in a high impact journal like Nature Comms.

Reviewer #5 (Remarks to the Author):

The authors have not responded satisfactory to my most crucial comment, namely point 4. The electron micrographs they provide confirm that the vesicles they have gated by FACS are too small to represent endosomes, with a diameter less than 10% of bona fide endosomes. In my comment I requested "high-resolution fluorescence microscopy (with appropriate quantifications) which should demonstrate clearly whether MAC and ZFYVE21 localize to Rab5 positive endosomes, and whether this depends on PI(3)P. The authors have made no attempt to perform this key analysis, and I thus cannot recommend publication.

Our specific responses to the reviewers are as follows:

Reviewer #2 (Remarks to the Author):

This is the third time I have reviewed this manuscript. I will restrict my comments to the responses (and changes) made in addressing my last review critique:

1. I asked for clarification regarding the dose of C9 used, the amount of label incorporated and whether there was significant aggregation of the C9 post-labelling given the propensity of this protein to aggregate. The majority of the response given does not address these questions. No analysis of labelling density or efficiency is made, instead the authors refer to "manufacturer's instructions" - this is not at all helpful. No attempt is made to test whether the C9 is aggregated which would not only reduce its function but also increase the likelihood of "non-specific" uptake into the cells. I do not understand why the authors cannot address the points made.

The dose of C9 used was 800ug per labeling reaction. 75uL of C9 after the labeling reaction was added to HUVEC in a total volume of 2000uL, meaning that the dose HUVEC were treated with 30ug of C9 AF647. We analyzed the labeling efficiency prior to use of three separate batches of C9AF647 was 0.52-1.21 moles of AF647 per mole C9 protein. The formula for calculating the efficiencies is: Moles dye pre mole protein= $A_{650} \times \text{dilution factor} / 239,000 \times \text{protein concentration (M)}$. This information has been added to the Methods

We performed new experiments to rule out the possibility that aggregation of labeled C9 protein, rather than incorporation of labeled C9 into MAC, was responsible for the uptake and signaling we have observed. The following text and figures were added:

“C9 has strong propensity to form aggregates and as such, FACS-sorted C9+Rab5+ events may have reflected structures containing aggregates of C9 rather than MAC. To test for C9 AF647 aggregation, we tested absorbance of C9 AF647 prior to and following ultracentrifugation of the protein product based on the reasoning that if significant protein aggregation occurred, we would expect that the measured absorbance of C9 AF647 supernatants to decrease following ultracentrifugation. We tested two separate batches of C9 AF 647 along with a commercial goat antimouse IgG AF647 Ab as a labeled protein control. Our data showed that the absorbance of the supernatants from two separate batches of C9 AF647 declined by $\leq 5\%$ after ultracentrifugation, which was less than that seen in the control (Fig S1d, left, middle). Changes in absorbance after ultracentrifugation did not differ between C9 AF647 batches and were not significantly different than changes in measured absorbance in

phosphate buffered saline (PBS) controls (Fig S1d, right). We subsequently tested the ability of ultracentrifugation products of C9 AF647 to be internalized into C9+Rab5+ endosomes. C9 AF647 supernatants prior to (C9 AF647 Pre) and following (C9 AF647 Post) centrifugation were used to treat HUVEC under complement-activating conditions. Our data showed that C9 AF647 Pre and C9 AF647 Post supernatants did not significantly differ in their ability to generate C9+Rab5+ vesicles in HUVEC (Fig S1e, bottom graph). Moreover, both C9 AF647 Pre and C9 AF647 Post products yielded significantly fewer C9+Rab5+ vesicles per cell when simply incubated with the cells compared to uptake of the same labeled proteins when combined with PRA sera. Together, these data indicated that C9 AF647 aggregates minimally contributed to the overall numbers of FACS-sorted C9+Rab5+ vesicles used for downstream proteomic analyses.”

Furthermore, treatment of HUVEC with C9AF647 in the absence of PRA did not induce ZFYVE21. These new data are now included in Fig 1c (right):

2. I asked why the authors did not here use C6- and/or C9-depleted/deficient sera (and addbacks) as controls, and ask for clarification that the MAC pore includes multiple copies of C9. The first question is ignored, the second addressed by a list of papers purported to support the contention that C5b-8 is a pore. The structural composition of the MAC pore is made very clear from recent structural reports. C5b-8 may disrupt lipid in some context but it CANNOT form a pore. This needs to be clear in the text to avoid propagating misunderstandings.

We have performed an additional *in vitro* experiment using C9-depleted sera with addback controls (lanes 8 vs 9) as requested by the reviewer. This study also demonstrates that C9AF647 alone does not induce ZFYVE21. This blot is shown in Fig 1c (right) as above.

After further reviewing the literature we believe that the reviewer is correct in his/her assertion that C5b-8 may disrupt lipid bilayers but cannot form a pore, and we thank the reviewer for pointing out this recent data. The following sentence has been added to the text on page 4: 'Complement components C5b-8 form a complex that is nonpore-forming until bound to polymeric C9 to form MAC.'

3. I asked why the authors do not test *in vivo* in deficient mice (that they wrongly stated were not available). Most of the response is irrelevant to the question asked. Only in the last two sentences is the question addressed with the statement that "In the future studies using C6 or C9 immunodeficient mice will be required to definitively show that the presently studied pathway is MACdependent these studies are outside the scope of the current manuscript". As I stated before, there is really no excuse for not testing in one of these lines. This is particularly the case for publication in a high impact journal like Nature Comms.

We wish to clarify a key misunderstanding re the use of existing complement-deficient mice. All of our *in vivo* experiments are based on studying human endothelium in order to avoid the critical species differences in the immunological functions of endothelial cells. We use female SCID/bg mice as hosts for human cells and tissues because they have mutations that prevent the development of T cells and that render NK cells dysfunctional. Both mutations are required for the animals to accept human vascularized and hematopoietic cell xenografts. The reviewer is correct in pointing out that C6 and C9 immunodeficient mice exist, but these mutations are not available on a SCID/bg background. The *in vivo* experiments reported in our study cannot be conducted without creating new mouse strains that would introduce complement gene mutations into SCID/bg mice. In the original decision letter, we were explicitly informed by the editors that we were not required to perform the suggested *in vivo* experiments in a revised ms. We hope this clarification satisfies Reviewer 3.

Reviewer #5 (Remarks to the Author):

The authors have not responded satisfactory to my most crucial comment, namely point 4. The electron micrographs they provide confirm that the vesicles they have gated by FACS are too small to represent endosomes, with a diameter less than 10% of bona fide endosomes. In my comment I requested "high-resolution fluorescence microscopy (with appropriate quantifications) which should demonstrate clearly whether MAC and ZFYVE21 localize to Rab5 positive endosomes, and whether this depends on PI(3)P. The authors have made no attempt to perform this key analysis, and I thus cannot recommend publication.

The isolated vesicles we used for proteomics are indeed too small to qualify as intact endosomes. As stated previously, we used this material for hypothesis generation by proteomic analyses and then validated these findings by immunostaining of intact cells. We based our assertion that the Rab5+ vesicles

that contain MAC were appropriate in size to be early endosomes based on immuno-electron microscopy. For this revision, we have performed 2 additional experiments to address the reviewer's concerns. In the first new experiment, we have performed 3 color confocal microscopy studies with quantification to demonstrate that ZFYVE21 colocalizes with C9+Rab5+ vesicles in HUVEC following PRA treatment (new Fig 1f). In the second experiment we performed confocal and super resolution fluorescence microscopy studies with quantification to demonstrate that PI(3)P lipid depletion attenuates formation of ZFYVE21+Rab5+ vesicles (new Fig 1k).

We believe that we have satisfactorily addressed the remaining concerns of both Reviewer 3 and Reviewer 5 and hope that the revised ms. is now acceptable for publication in Nature Communications.

Sincerely,

Dan Jane-wit

Reviewers' comments:

Reviewer #5 (Remarks to the Author):

The authors have now performed the microscopy experiments I requested. However, I still have a couple of concerns:

1. It appears that there is indeed recruitment of ZFYVE21 to Rab5-positive endosomes upon PRA incubation, and that treatment with PIKIII inhibitor abolishes this co-localization. However, one would have predicted that ZFYVE21 became cytosolic in the absence of endosomal PI3P, but instead it can be found on a separate vesicle population. Which are these vesicles, and how do the authors explain this relocalization?
2. The super-resolution microscopy experiments are poorly described, but presumably they represent STED microscopy since this is mentioned in Materials and Methods. In any case, it seems meaningless to use super-resolution microscopy as basis for calculating intermolecular distances, especially as these are calculated to be very large even in the presence of PRA and absence of PIKIII inhibitor.

Reviewer #6 - replacement for R#2 (Remarks to the Author):

Following your request to judge the response of the authors to Reviewer #2, I went over the rebuttal letter and examined briefly the text and data.

Here are a few comments:

1. This team has published 2 earlier manuscripts, one in PNAS and one in Circulation, on the consequences of activation of non-canonical NF-kappa B in endothelial cells by complement C5b-9. In this manuscript they further characterize this signalling and implicate ZFYVE21 in the activation.

2. I expected them to give credits in their manuscript to earlier manuscripts dealing with non-lytic stimulatory effects of C5b-9; but did not find it.

3. In response to the first comment of Reviewer #2, they added information on the C9 labelling conditions and employed ultracentrifugation of the labelled C9 to show that it did not undergo aggregation during its labelling. Although the reviewer did not ask for that, I would have expected them to test the labeled C9 in a hemolytic titration assay relative to native C9.

4. In response to the second comment of Reviewer #2, they added a sentence (line 78) : Complement components C5b-8 form a complex that is nonpore-forming until bound to polymeric C9 to form MAC. Strangely enough, this sentence is in the Results chapter, the finding was shown in several much earlier papers by others, but no reference is given. BTW, the correct finding was that "C5b-8 initiates polymerization of C9" rather than "C5b-8 ... binds to to polymeric C9".

5. Also, in response to the second comment of Reviewer #2, they added a figure (1c, right) and in the legend (line 867) they write: HUVEC were treated with IgG+ and IgG- sera fractions from PRA, C9-deficient sera, and C9 AF647 as indicated for 30 min prior to Western blotting of whole cell lysates (C, right). This does not fit with the lanes shown in the figure.

6. In response to the third comment of Reviewer #2, the authors rebut the claim of the reviewer that their findings must be confirmed in vivo in a fully murine model system. I support the claim of the reviewer. Using human cells and human serum in the mouse background may generate artifactual activations.

7. Additional comment: too many of the claims in this manuscript are based on co-localizations in low-resolution confocal fluorescence images.

Reviewers' comments:

Reviewer #5 (Remarks to the Author):

The authors have now performed the microscopy experiments I requested. However, I still have a couple of concerns:

1. It appears that there is indeed recruitment of ZFYVE21 to Rab5-positive endosomes upon PRA incubation, and that treatment with PIKIII inhibitor abolishes this co-localization. However, one would have predicted that ZFYVE21 became cytosolic in the absence of endosomal PI3P, but instead it can be found on a separate vesicle population. Which are these vesicles, and how do the authors explain this relocalization? **This is an interesting observation, and we have not comprehensively studied the identity of these vesicles. We expect that blocking ZFYVE21 colocalization with Rab5+ endosomes would result in targeting of this molecule towards degradative pathways as the inability of ZFYVE21 to interact with Rab5 in cells containing Rab5 dominant negative constructs (Fig 1e) or in cells depleted of PI(3)P with KU-55933 (Fig 1h,i) or MTM1 overexpression (Fig 1j) profoundly reduced the overall stability of ZFYVE21. As such, blocking ZFYVE21 interaction with Rab5 likely shunted this molecule to vesicles implicated in protein degradation such as autophagolysosomes. The following text has been added to the Results section:**

In these studies, PIKIII, redistributed ZFYVE21 to a distinct Rab5- vesicular population rather than into the cytosol. Given our prior observations of reduced stability of ZFYVE21 in cells containing Rab5 DN (Fig 1e) and in cells depleted of PI(3)P (Fig 1h,i,j), we surmised that ZFYVE21 was shunted to vesicles implicated in protein degradation .when interactions between ZFYVE21 and Rab5 were blocked.

2. The super-resolution microscopy experiments are poorly described, but presumably they represent STED microscopy since this is mentioned in Materials and Methods. In any case, it seems meaningless to use super-resolution microscopy as basis for calculating intermolecular distances, especially as these are calculated to be very large even in the presence of PRA and absence of PIKIII inhibitor. **STED imaging was indeed used for the super-resolution microscopy studies, and a more thorough explanation of how these experiments were performed are provided in the Materials & Methods section. The STED imaging was employed as a correlate modality to the confocal microscopy studies shown in the same figure showing colocalization of ZFYVE21 and Rab5 and for Western blot co-IP data showing interaction between Rab5 and ZFYVE21.**

In the STED imaging studies, the distances between ZFYVE21 and Rab5 were significantly increased in the presence of the PIKIII inhibitor which we interpreted as a lack of

colocalization following depletion of PI(3)P. This finding supported the data derived from parallel confocal microscopy experiments in the same figure showing decreased numbers of Rab5+ZFYVE21+ endosomes following PI(3)P lipid depletion and various Western blot data showing that blocking Rab5-ZFYVE21 interactions with PI(3)P lipid depletion decreased ZFYVE21 protein stability.

Reviewer #6 - replacement for R#2 (Remarks to the Author):

Following your request to judge the response of the authors to Reviewer #2, I went over the rebuttal letter and examined briefly the text and data.

Here are a few comments:

1. This team has published 2 earlier manuscripts, one in PNAS and one in Circulation, on the consequences of activation of non-canonical NF-kappa B in endothelial cells by complement C5b-9. In this manuscript they further characterize this signaling and implicate ZFYVE21 in the activation.
2. I expected them to give credits in their manuscript to earlier manuscripts dealing with non-lytic stimulatory effects of C5b-9; but did not find it. **The appropriate citations have been added to the manuscript.**
3. In response to the first comment of Reviewer #2, they added information on the C9 labelling conditions and employed ultracentrifugation of the labelled C9 to show that it did not undergo aggregation during its labelling. Although the reviewer did not ask for that, I would have expected them to test the labeled C9 in a hemolytic titration assay relative to native C9. **Based on our prior and current work, MAC deposition on nucleated endothelial cells with PRA sera does not cause significant cytolysis, which was supported by the work of others, albeit using different models of MAC assembly on different cell types (references provided by the reviewer in point #2). As such we did not use C9 AF647 to study complement-induced hemolysis but instead we used this reagent as a tool to identify new signaling components relevant to MAC-induced activation of non-canonical NF-kB. To this end, we were able to recapitulate salient features of non-canonical NF-kB with this reagent, identifying in an unbiased fashion ZFYVE21 as a mediator of this process. We agree that future studies examining the hemolytic activity of PRA sera in anucleated cells like RBCs would warrant hemolytic titration assays. As such, the handling editor has not mandated that we perform this experiment for the present study.**

4. In response to the second comment of Reviewer #2, they added a sentence (line 78) : Complement components C5b-8 form a complex that is nonpore-forming until bound to polymeric C9 to form MAC. Strangely enough, this sentence is in the Results chapter, the finding was shown in several much earlier papers by others, but no reference is given. BTW, the correct finding was that "C5b-8 initiates polymerization of C9" rather than "C5b-8 ... binds to to polymeric C9". **The sentence has been revised and moved to the Results section, where references are provided per the reviewer's request. The sentence now reads:**

MAC are formed from five terminal complement components and intercalate upon assembly into target cell membranes to induce various non-cytolytic features⁷. Complement components C5b-8 form a complex that is nonpore-forming until initiation of polymerization of C9 by this same complex to form MAC.

5. Also, in response to the second comment of Reviewer #2, they added a figure (1c, right) and in the legend (line 867) they write: HUVEC were treated with IgG+ and IgG- sera fractions from PRA, C9-deficient sera, and C9 AF647 as indicated for 30 min prior to Western blotting of whole cell lysates (C, right). This does not fit with the lanes shown in the figure. **We have changed the wording in the figure legend in an effort to improve its clarity as per the reviewer's request.**

6. In response to the third comment of Reviewer #2, the authors rebut the claim of the reviewer that their findings must be confirmed in vivo in a fully murine model system. I support the claim of the reviewer. Using human cells and human serum in the mouse background may generate artifactual activations. **We have not observed increased levels of C4d, NIK, or E-selectin in human artery segments in control SCID/bg injected with the IgG-fraction of PRA sera in prior studies, ruling out a soluble contaminant causing artifactual EC activation. In the current study, we similarly did not observe an induction of ZFYVE21 or SMURF2 with IgG- PRA sera (Fig 5 and Fig S3). Additionally, we have provided *in vitro* data showing that MAC and not IgG or anaphylatoxins or C9 AF647 aggregates activated ZFYVE21 in EC (Fig 1c).**

We do not rebut the importance of using murine models, but wish to reiterate that humanized model systems were chosen so that we would be able to address signaling mechanisms and phenotypes associated with direct allorecognition. Given the above and our numerous experimental and verbal responses to the prior reviewer, we have not been mandated by the handling editor to perform further experiments related to the use of murine systems.

7. Additional comment: too many of the claims in this manuscript are based on co-localizations in low-resolution confocal fluorescence images. Colocalization of key molecules in the study including ZFYVE21, Rab5, PTEN, and NIK were shown using proteomic data of FACS-sorted endosomes, Western blot data, enzymatic studies in the case of PTEN, and in the case of ZFYVE21 and Rab5, STED imaging. Taken together, these multiple approaches altogether strengthened our claims made from confocal imaging alone.